# Two-dimensional organic-inorganic hybrid perovskite quantum-well nanowires enabled by directional noncovalent intermolecular interactions

Meng Zhang[1,2], Leyang Jin[1,2], Tianhao Zhang[1], Xiaofan Jiang ©[1], Mingyuan Li[1], Yan Guan[1] & Yongping Fu ©[1] ✉

Layered 2D semiconductors, when grown into 1D nanowires, can exhibit excellent optical and electronic properties, promising for nanoscale optoelectronics and photonics. However, rational strategies to grow such nanowires are lacking. Here, we present a large family of quantum-well nanowires made from 2D organic-inorganic hybrid metal halide perovskites with tunable well thickness, organic spacer cations, halide anions, and metal cations, achieved by harnessing directional nonvalent intermolecular interactions present among certain spacer cations. The unusual 1D anisotropic growth within the 2D plane is induced by preferential self-assembly of selected spacer cations along the direction of stronger intermolecular interactions and further promoted by crystal growth engineering. Owing to the intrinsic 2D quantum-well-like crystal structures and 1D photon confinement at the subwavelength scale, these nanowires exhibit robust exciton-photon coupling, with Rabi splitting energies of up to 700 meV, as well as wavelength-tunable and more efficient lasing compared to exfoliated crystals.

Semiconductor nanowires (NWs) exhibit intriguing 1D crystal growth and potential applications in integrated optoelectronic and photonic circuits[1–5]. These NWs are generally derived from semiconductors with three-dimensional (3D) crystal structures, including traditional inorganic semiconductors and emerging halide perovskites. Particularly, 3D perovskite NWs have recently been recognized for their strong light-matter interactions, facilitating the studies of exciton-polaritons (i.e., a hybrid quasiparticle resulting from strong coupling of exciton and cavity photon) and offering prospects in lasing and quantum optics[6–9]. However, a fundamental limitation is their relatively low exciton binding energies, insufficient to prevent exciton dissociation at high carrier densities. 2D organic-inorganic layered perovskites, which are essentially self-assembled quantum wells of 3D perovskites, present a paradigm for NW application[10–12]. These

materials can exhibit higher photoluminescence (PL) quantum yields, larger exciton binding energies, and tunable excitonic properties through chemical tailoring, making them more attractive in applications such as exciton-polariton devices and light-emitting devices[3,7,14]. It will be of interest to incorporate these inherent quantum-well structures in the nanowire morphology, where the lateral confinement could further provide an additional tuning knob for their properties, as demonstrated by multi-quantum-well NW heterostructures based on conventional inorganic semiconductors[15–17].

The 1D anisotropic NW growth of 3D semiconductors has been enabled by vapor-liquid-solid (VLS) and screw dislocation mechanisms[1,18–20], extending these methods to 2D perovskite NWs remains challenging due to their dominant crystal growth morphology determined by their intrinsic 2D crystal structures[10,21]. The VLS growth

[1]Beijing National Laboratory for Molecular Science, College of Chemistry and Molecular Engineering, Peking University, Beijing 100871, China. [2]These authors contributed equally: Meng Zhang, Leyang Jin. ✉e-mail: yfu@pku.edu.cn

method requires a metal catalyst droplet to initiate 1D anisotropic crystal growth. However, finding such a catalyst for 2D perovskites is unlikely. NWs may form via the propagation of axial screw dislocation[20], a line defect that shears part of the crystal lattice in a specific direction. In 2D layered materials, this shear typically occurs along the stacking direction, leading to a spiral growth[22]. This particular growth mode, combined with the catalyzed VLS method, has enabled NWs growing along the stacking direction, as observed in a special case of layered GeS NWs[23,24]. However, most 2D layered materials, in the absence of a catalyst droplet, tend to grow slower in the stacking direction compared to the in-plane direction, resulting in the formation of 2D plate-like structures rather than 1D NWs[25,26].

NWs of several 2D perovskites have been synthesized by vapor phase deposition and templated growth[27,28], but the organic spacer cations used have been limited to butylammonium, and these methods are difficult to obtain phase-pure 2D perovskites with $n > 2$, where $n$ refers to the thickness of the inorganic layers in units of metal halide octahedra. Recently, Shao et.al have synthesized 2D perovskite NWs through molecular templating of the spacer cations[29], and the 1D anisotropic growth is attributed to the solvation effects of carboxylic acids present in the specially designed spacer cations. These NWs exhibit anisotropic emission polarization, low waveguiding loss coefficients, and efficient light amplification. Here, we report an alternative, universal mechanism for the formation of single-crystal NWs of 2D perovskite NWs and strong light-matter interactions from these NWs. Unlike the in-plane 2D growth observed in conventional 2D layered materials, we find certain 2D perovskites can exhibit significant 1D anisotropic growth driven by directional noncovalent intermolecular interactions among organic spacer cations. This inherent crystal growth habit can be harnessed to enable the growth of NWs with high aspect ratios (>200). By exploring various spacer cations with targeted intermolecular interactions, we synthesize single-crystal NWs of

diverse 2D perovskite quantum-wells (up to 21 phases), including those with varying inorganic layer thickness and metal cations.

## Results

### Crystal growth habits of two-dimensional perovskites

We first present the influence of the spacer cations on the crystal growth behaviors of the simplest 2D perovskites with $n = 1$, $(LA)_2PbI_4$, where $LA^+$ is a large organic ammonium spacer cation. These structures are formed by the van der Waals stacking of a single corner-sharing $PbI_6^{4-}$ octahedral layer sandwiched by two self-assembled layers of organic spacer cations (Fig. 1a). Figure 1b shows the chemical structures of a range of spacer cations, including both aliphatic and aromatic ammonium cations. Typically, aqueous solution growth of $(LA)_2PbI_4$ lead to distinct macroscopic shapes of the crystals, most commonly, 2D plate-like (Fig. 1c), as intuitively understood due to their intrinsic layered crystal structures and previously observed. For example, common spacers cations such as butylammonium ($BA^+$), hexylammmonium ($HA^+$), phenylethylammonium ($PEA^+$), and 4-trifluoromethylphenylmethylammonium ($4CF_3PMA^+$) predominantly form more plate-like crystals. Sometimes, rectangular crystal shapes of $(HA)_2PbI_4$ and $(BA)_2PbI_4$ can be observed, indicating minor anisotropy, as will be further discussed later. However, some 2D perovskites naturally grown into 1D ribbon-like or needle-like morphologies (Fig. 1d), which are significantly less common. By examining the crystal structures in the crystallographic database, we found spacers cations such as phenylmethylammonium ($PMA^+$), 4-fluorophenylmethylammonium ($4FPMA^+$), 2-fluorophenylmethylammonium ($2FPMA^+$), 4,4-difluoropiperidinium ($DFPD^+$), 2-trifluoromethylphenylethylammonium ($2CF_3PEA^+$), 2-bromophenylethylammonium ($2BrPEA^+$), methylbenzylammonium ($MBA^+$), pentylammonium ($PA^+$), 4-ammoniumbutyric acid ($ABA^+$), and 3,3-difluoropyrrole ammonium ($DFP^+$) tend to facilitate the formation of elongated or needle-like crystals.

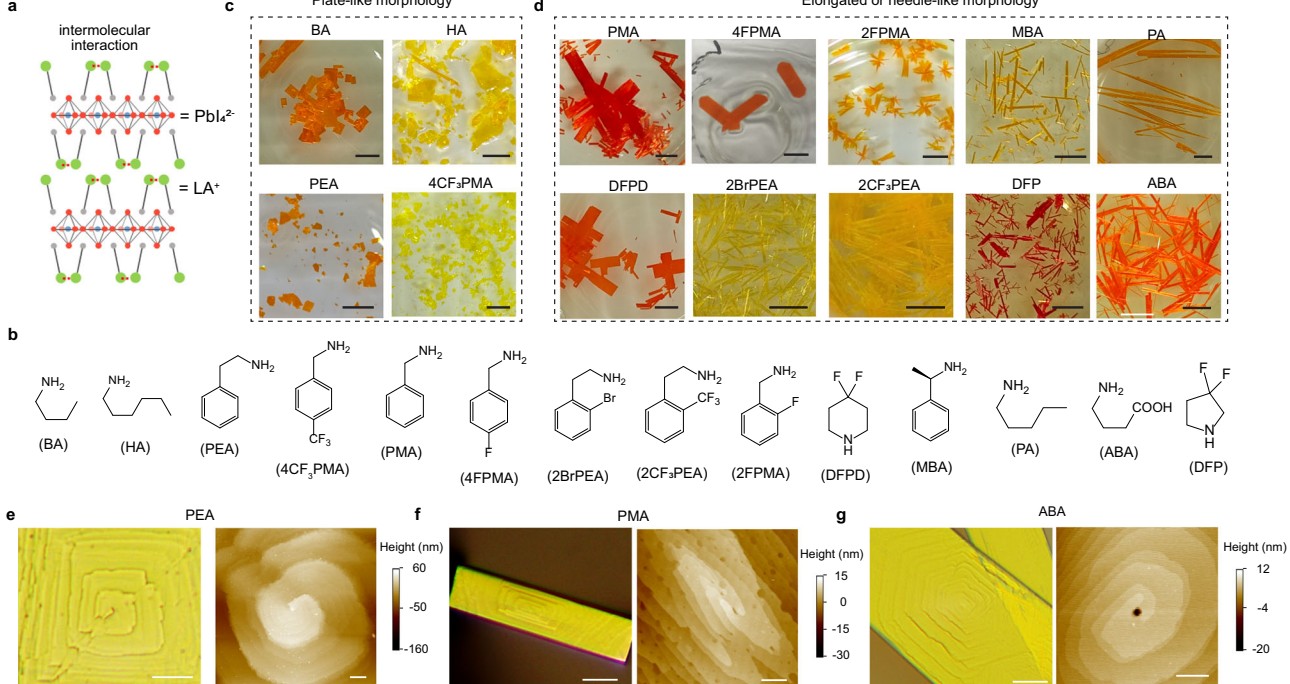

**Fig. 1 | Crystal growth habits of 2D perovskites. a** Schematic crystal structure of 2D perovskite $(LA)_2PbI_4$, where $LA^+$ is a large organic ammonium cation. **b**, Molecular structures of various spacer cations. **c**, **d** Optical images of various 2D perovskite crystals, showing more plate-like morphologies (**c**) and more needle-like morphologies (**d**). All scale bars are 2.7 mm. **e**–**g** Optical and atomic force microscope (AFM) images of $(PEA)_2PbI_4$ (**e**), $(PMA)_2PbI_4$ (**f**), and $(ABA)_2PbI_4$ (**g**) microcrystals showing the screw dislocation cores. Scale bars in the optical images are 20 μm. Scale bars in the AFM images are 2 μm, 500 nm, and 500 nm, respectively.

We then employed in-situ optical microscopy to monitor the crystal growth process. The experiment involved placing a droplet of a slightly warm saturated perovskite solution onto a glass slide. As the solution cooled and the solvent evaporated, it transitioned into a supersaturation state, triggering nucleation and subsequent the growth of various microstructures. The observed morphologies were consistent with the forms seen in their corresponding macroscopic crystals (Supplementary Fig. 1). In addition, many objects exhibit screw dislocation cores on the surfaces, indicating that dislocation-driven growth mode[25] is the primary growth mechanism for the 2D perovskites under our experimental conditions. Atomic force microscopy (AFM) on 2D plates such as (PEA)$_2$PbI$_4$ (Fig. 1e) reveal nearly isotropic dislocation cores, characterized by the circular or square shapes of the terraces, indicative of similar growth rates in the two orthogonal directions within the 2D plane. By contrast, elongated (needle-like or ribbon-like) structures such as (PMA)$_2$PbI$_4$ (Fig. 1f) and (ABA)$_2$PbI$_4$ (Fig. 1g) display significantly anisotropic dislocation cores, with a more rapid expansion rate in in the longer axis of the object compared to the shorter axis. More spiral structures are provided in Supplementary

Fig. 2. In addition, we observed that when a hotter precursor solution is dispensed onto the glass substrate, some 2D perovskites, such as (PMA)$_2$PbI$_4$ and (MBA)$_2$PbI$_4$, could from a cross-shaped morphology, exhibiting twining at the branched center (Supplementary Fig. 3).

## Anisotropic growth induced by intermolecular interactions

To understand the crystal growth mechanism driving 1D macroscopic crystal shapes, we conducted single-crystal X-ray diffraction (SCXRD) experiments on the elongated crystals to identify the growth direction. Using (PMA)$_2$PbI$_4$ as an example (see crystal structure shown in Fig. 2a), the SCXRD results show the longer crystal direction aligns with the in-plane $b$-axis and the six exposed faces of the crystal correspond to the (100), (010), and (001) planes (Fig. 2b). There are two distinct orientations of PMA$^+$ cations within each self-assembled layer, arranged in an edge-to-face configuration (Fig. 2c), which is stabilized by C-H···π interactions between aromatic C-H groups and π-systems (i.e., a weak hydrogen bond)[30]. Along the $b$-axis, the proximity between the hydrogen atom and the π-system ($d_{H-pi}$ = 2.9 Å) of two neighboring PMA$^+$ cations approach the sum of their van der Waals radii (2.8 Å),

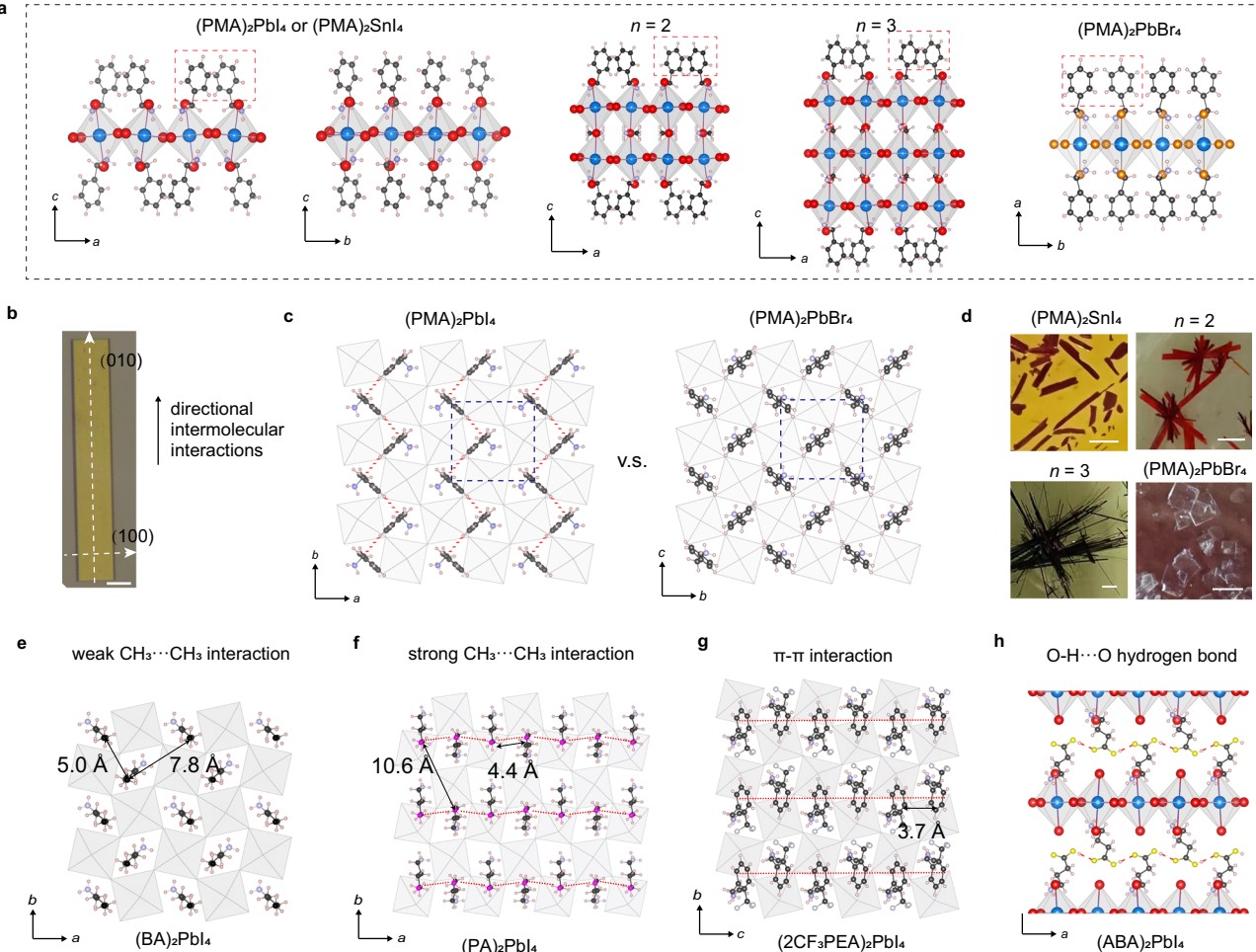

**Fig. 2 | Mechanisms of in-plane anisotropic growth of 2D perovskites driven by directional noncovalent intermolecular interactions. a** Crystal structures of (PMA)$_2$PbI$_4$ or (PMA)$_2$SnI$_4$ viewed along the $b$- and $a$-axis, (PMA)$_2$(MA)$_{n-1}$Pb$_n$I$_{3n+1}$ ($n$ = 2 and 3), and (PMA)$_2$PbBr$_4$. **b** Optical image of a (PMA)$_2$PbI$_4$ crystal, showing the longer axis which corresponds to (010) direction. The scale bar is 20 μm. **c** The arrangement of the PMA$^+$ cations in (PMA)$_2$PbI$_4$ and (PMA)$_2$PbBr$_4$ viewed along the out-of-plane direction, showing directional C-H···π interactions along the $b$-axis in (PMA)$_2$PbI$_4$ (red dashed lines). **d** Optical images of (PMA)$_2$SnI$_4$, (PMA)$_2$(MA)$_{n-1}$Pb$_n$I$_{3n+1}$ ($n$ = 2, or 3), and (PMA)$_2$PbBr$_4$ crystals. (PMA)$_2$PbBr$_4$ forms square-flake crystals, whereas the other three form elongated needle-like crystals.

All scale bars are 1.25 mm. **e** The arrangement of the BA$^+$ cations in (BA)$_2$PbI$_4$ viewed along the out-of-plane direction, showing weak CH$_3$···CH$_3$ van der Waals interactions. **f** The arrangement of the PA$^+$ cations in (PA)$_2$PbI$_4$ viewed along the out-of-plane direction, showing directional CH$_3$···CH$_3$ van der Waals interactions along the $a$-axis (red dashed lines). **g** The arrangement of the 2CF$_3$PEA$^+$ cations in (2CF$_3$PEA)$_2$PbI$_4$ viewed along the $a$-axis, showing directional π···π stacking interactions along the $c$-axis (red dashed lines). **h**, The arrangement of the ABA$^+$ cations in (ABA)$_2$PbI$_4$ viewed along the $b$-axis, showing directional hydrogen bonding interactions along the $a$-axis (red dashed lines).

indicating a strong C-H···π interaction that leads to the formation of a directional molecular chain. In contrast, the distances among the adjacent PMA$^+$ cations along the $a$-axis are 2.9 Å and 5.1 Å, the latter suggesting a lack of C-H···π interaction chain. The directional C-H···π interactions thus facilitate the self-assembly of PMA$^+$ cations along the $b$-axis, which results in the anisotropic dislocation cores shown in Fig. 1f. Similar results were obtained for (MBA)$_2$PbI$_4$ (Supplementary Fig. 4). In addition, to determine the growth direction of the cross-shaped crystals, we cleaved the branches and performed SCXRD analysis on each (Supplementary Fig. 3). The results show that each branch follows the same crystallographic direction, corresponding to stronger intermolecular interactions.

To further confirm the role and the generality of anisotropic directional C-H···π interactions in inducing 1D growth, we explored the homologous series of 2D Ruddlesden-Popper (RP) perovskites, (PMA)$_2$(MA)$_{n-1}$Pb$_n$I$_{3n+1}$ (MA$^+$ = methylammonium), with $n$ value indicating the inorganic well thickness, as well as (PMA)$_2$SnI$_4$ and (PMA)$_2$PbBr$_4$ (Fig. 2a). (PMA)$_2$(MA)Pb$_2$I$_7$, (PMA)$_2$(MA)$_2$Pb$_3$I$_{10}$, and (PMA)$_2$SnI$_4$ display the same packing arrangement of the PMA$^+$ cations as observed in (PMA)$_2$PbI$_4$. As expected, these three structures crystallize into needle-like crystals (Fig. 2d). In contrast, (PMA)$_2$PbBr$_4$ adopts 2D plate-like morphology.

Analyses of the crystal structures reveal that the benzene rings of PMA$^+$ cations in (PMA)$_2$PbBr$_4$ exhibit nearly isotropic intermolecular interactions, occupying the center of the pockets formed by four lead bromide octahedra. The distances between neighboring PMA cations are uniform, and each PMA$^+$ forms weak C−H···π interactions (~3.7 Å) with four adjacent cations, leading to isotropic intermolecular interactions along the two in-plane directions (Fig. 2c). In contrast, in (PMA)$_2$PbI$_4$, the packing distances of the PMA$^+$ cations increase to accommodate the expanded inorganic lattice, as the lead iodide framework provides a larger space for the PMA$^+$ cations. This expansion causes the PMA$^+$ cations to displace from the center, forming two strong C−H···π interactions along the $b$-axis. This structural change is reminiscent of the phenomena observed in 3D perovskites[31], where an undersized A-site cations, such as methylammonium (MA$^+$) and Cs$^+$ in MAPbBr$_3$ and CsPbBr$_3$, induces octahedral tilt and A-cation off-centering displacement when deceasing temperature. A similar comparison can be drawn between cubic MAPbBr$_3$ perovskite and tetragonal MAPbI$_3$ perovskite[32–34]. Lead and tin cations exhibit similar ionic radius, and therefore the intermolecular interactions of the spacer cations in the analogous Pb and Sn 2D perovskites structures are typically similar.

Incorporation of halogen or alkyl groups at the 4-postion of the PMA$^+$ cation induces halogen bonds or steric effect, which could affect the molecular packing configurations and potentially disrupt the C-H···π interactions between the adjacent cations. For example, while (PMA)$_2$PbI$_4$ and (4FPMA)$_2$PbI$_4$ exhibit needle-like crystals, compounds such as (4ClPMA)$_2$PbI$_4$ (4ClPMA$^+$ = 4-chlorophenylmethylammonium) and (4CF$_3$PMA)$_2$PbI$_4$ form 2D plate-like morphologies. The halogen substitution introduces halogen-π interactions among the interlayer spacer cations[35]. The strength of these interactions depends on the size and depth of the electrophilic region (the σ-hole) associated with the halogen atom, which follows the trend: I > Br > Cl ≫ F[36]. In (4ClPMA)$_2$PbI$_4$, the halogen-π interactions among 4ClPMA$^+$ cations across layers disturb the edge-to-face arrangement within layers, leading to a more parallel face-to-face packing arrangement and thereby interrupting the directional C-H···π interactions network (Supplementary Fig. 5). In contrast, the halogen-π interactions in 4FPMA$^+$ are significantly weaker than in 4ClPMA$^+$, allowing the edge-to-face configuration and C-H···π interactions to persist in the former. Incorporating -CF$_3$ group can also alter the edge-to-face configuration of the PMA$^+$ cations. While the benzene rings of the 4CF$_3$PMA$^+$ cations are slightly displaced from the center (Supplementary Fig. 6), the absence of a robust and directional C-H···π interactions chain, as observed in (PMA)$_2$PbI$_4$, results in the 2D growth of (4CF$_3$PMA)$_2$PbI$_4$.

Beyond C-H···π interactions, we found that other types of directional intermolecular interactions, such as van der Waals interactions (CH$_3$···CH$_3$), aromatic π···π stacking, and strong hydrogen bonds, could also be utilized to induce 1D anisotropic growth of 2D perovskites. Among these, van der Waals CH$_3$···CH$_3$ interactions are the weakest[30] and are commonly observed in 2D alkylammonium lead iodide perovskites. To investigate their influence, we studied the crystal growth behaviors of (BA)$_2$PbI$_4$, (PA)$_2$PbI$_4$, and (HA)$_2$PbI$_4$. Due to the undersized nature of alkylammonium cations relative to the lead iodide framework, the tails of these cations typically exhibit off-centering displacements in the crystal structures, forming directional and anisotropic intermolecular interactions. For example, the crystal structure of (BA)$_2$PbI$_4$ exhibits two distinct arrangements of BA$^+$ cations, with 5.0 Å between the neighboring -CH$_3$ groups along the $b$-axis, compared to distances of 5.0 Å and 7.8 Å along the $a$-axis (Fig. 2e). Under low-supersaturation growth conditions, minor anisotropic growth can be observed by the rectangular crystal shapes of some (BA)$_2$PbI$_4$ and (HA)$_2$PbI$_4$ (Supplementary Fig. 7). However, the relatively weak anisotropic van der Waals interactions in (BA)$_2$PbI$_4$ and (HA)$_2$PbI$_4$ are not sufficient to drive significant 1D growth. In contrast, (PA)$_2$PbI$_4$ exhibits stronger directional van der Waals interactions. The distances between the neighboring -CH$_3$ groups along the $a$-axis are 4.4 Å, while the corresponding distances along the $b$-axis are 4.4 Å and 10.6 Å (Fig. 2f). The greater difference in the carbon-carbon distances between adjacent spacer cations along the two in-plane directions reflects stronger directional van der Waals interactions, which lead to 1D growth in in (PA)$_2$PbI$_4$.

The pronounced difference in the crystal growth behavior of (PA)$_2$PbI$_4$ compared to (BA)$_2$PbI$_4$ and (HA)$_2$PbI$_4$ arises from differences in their structural phases and the associated intermolecular interaction strength. At high temperatures above 330 K, the three structures are isostructural[37]. As the temperature decreases, the spacer cations deviate further from the center of the pockets, enhancing the directional intermolecular interactions that drive phase transitions (Supplementary Fig. 8). A key distinction of (PA)$_2$PbI$_4$ is its higher phase transition temperature 319 K, which is above the temperature used for nanowire growth. Consequently, the room temperature phase of (PA)$_2$PbI$_4$ is more closely related to the low-temperature phases of (BA)$_2$PbI$_4$ and (HA)$_2$PbI$_4$, which exhibit stronger anisotropic interactions. Supplementary Table 1 shows the discrepancies in the carbon-carbon distances between adjacent spacer cations along the two in-plane directions for the three structures. The smaller discrepancies in (BA)$_2$PbI$_4$ and (HA)$_2$PbI$_4$ indicates weaker anisotropic intermolecular interactions compared to (PA)$_2$PbI$_4$ in their room temperature phases.

To further highlight the impact of attenuated anisotropy leading to disappearance of 1D growth in (BA)$_2$PbI$_4$ and (HA)$_2$PbI$_4$, we performed crystal growth of (PA)$_2$PbI$_4$ at 340 K, above its phase transition. At this temperature, (PA)$_2$PbI$_4$ adopt a structure isostructural to the room temperature phases of (BA)$_2$PbI$_4$ and (HA)$_2$PbI$_4$, with longer distances between the tails of neighboring PA$^+$ cations compared to room-temperature phase (PA)$_2$PbI$_4$ (Supplementary Fig. 8 and Supplementary Table 1). As expected, (PA)$_2$PbI$_4$ grown under 340 K forms 2D plate-like crystals (Supplementary Fig. 9). These results suggest that the directional CH$_3$···CH$_3$ van der Waals interactions lack sufficient strength to drive 1D growth in the room-temperature phases of (BA)$_2$PbI$_4$ and (HA)$_2$PbI$_4$. However, these interactions can become significant when further enhanced, thereby promoting anisotropic growth in (PA)$_2$PbI$_4$.

The formation of aromatic π···π stacking between neighboring spacer cations is rarer than expected, despite many spacer cations having aromatic rings. This is because the aromatic rings are often too far apart to form π···π interaction (Supplementary Fig. 6). For example, in (PEA)$_2$PbI$_4$, the benzene rings of the PEA$^+$ cations are positioned at the center of the pockets formed by four lead iodide octahedra. The adjacent benzene rings exhibit a face-to-face configuration. There is

neither characteristic of C−H⋯π interactions, as seen in $(PMA)_2PbI_4$, nor any overlap of the adjacent rings, indicating an absence of π⋯π interactions. Consequently, there are no directional intermolecular interaction formed along the two in-plane directions, leading to 2D growth (Supplementary Fig. 6).Interestingly, substituents at 2-poisiton of the $PEA^+$ cation can introduce directional π⋯π interactions of the spacer cations. For example, in $(2CF_3PEA)_2PbI_4$, the benzene rings of the $2CF_3PEA^+$ cations exhibit significant off-centering displacements, forming strong directional π⋯π stacking between adjacent spacer cations along the $a$-axis, as characterized by a short inter-ring distance of 3.7 Å and partial overlapping of aromatic systems[38] (Fig. 2g and Supplementary Fig. 10). This is because that the incorporation of -$CF_3$ group introduces a steric effect, forcing the neighboring aromatic rings to offset opposite and form π⋯π interaction. A similar arrangement is observed in $(2BrPEA)_2PbI_4$, though the extent of π-system overlap is smaller, likely due to the lesser steric effect of the bromine atom compared to -$CF_3$ group (Supplementary Fig. 10). In both cases, the crystals grow faster along the π⋯π stacking direction.

The incorporation of carboxyl group in the spacer cations can introduce directional intermolecular hydrogen bond. For example, in $(ABA)_2PbI_4$, each $ABA^+$ cation is involved in hydrogen bonding with neighboring cations across the adjacent layer, forming a molecular chain structure along the $a$-axis, whereas such hydrogen bonding interactions are absent along the $b$-axis[39] (Fig. 2h). SCXRD analyses confirmed the growth direction along the $a$-axis (Supplementary Fig. 11). In the structures of $(DFP)_2PbI_4$ and $(DFPD)_2PbI_4$, the crystal growth directions were also observed to align with directional intra-layer intermolecular F⋯H hydrogen bond (Supplementary Fig. 12). Recently, spacer cations incorporating carboxyl groups, such as 2-(2,5-dicarboxyphenoxy)ethan-1-aminium ($TPA3^+$) and 2-(2-bromo-5-carboxyphenoxy)ethan-1-aminium ($BrCA3^+$), have been employed to synthesize 2D perovskite NWs, e.g., $(TPA3)_2PbBr_4$ and $(BrCA3)_2PbBr_4$[29]. The observed 1D anisotropic growth in these structures is attributed to the solvation effect of dangling carboxylic acids on the exposed crystal facets, which hinder crystal growth in these facets and promoting growth along a specific crystallographic direction. Interestingly, the 1D growth direction in $(TPA3)_2PbBr_4$ does not align with the directional hydrogen bond, indicating that their proposed growth mechanism differs from our mechanism, which utilizes directional intermolecular interactions to guide anisotropic growth. In addition to the solvation effects, we speculate that the presence of directional π⋯π stacking interactions in $(TPA3)_2PbBr_4$ and $(BrCA3)_2PbBr_4$ may also contribute to the 1D growth, as these interactions align with the observed 1D crystal growth directions in both structures (Supplementary Fig. 13). Due to the hydrophobic nature of the aromatic rings, the π⋯π interactions would dominate the self-assembly process of the spacer cations in the aqueous solution, similar to cases of $(2CF_3PEA)_2PbI_4$ and $(2BrPEA)_2PbI_4$.

The growth of hybrid 2D perovskites requires a synergy between the inorganic and organic lattices. The anisotropy of the inorganic lattices can be characterized by the difference between two in-plane lattice parameters, $D_1$ and $D_2$, which represent the distances between the second-nearest lead atoms (Fig. 3a, b, and Table 1). When spacer cations are positioned at the center of the pockets formed by four lead halide octahedra, the inorganic lattices are isotropic along the two in-plane directions, as observed in $(PEA)_2PbI_4$ and $(PMA)_2PbBr_4$. However, when the spacer cations are displaced from the center, directional intermolecular interactions and structural anisotropy of the inorganic lattice emerge, as observed in $(PMA)_2PbI_4$ and $(2CF_3PEA)_2PbI_4$. Furthermore, there is a strong correlation between directional intermolecular interactions and the anisotropy of the crystal structure. As shown in Table 1, the directional intermolecular interactions typically align with the more denser packing directions of the inorganic lattices.

The anisotropic shape and in-plane growth rates of the crystals can be effectively explained by the Wulff construction, which states that crystals preferentially expose facets with lower energy while facets with higher surface energy exhibit faster growth rates[40]. The overall surface energy is the combined contribution of both the organic and inorganic lattices. In the case of inorganic lattice, a denser atomic packing generally indicates stronger chemical bonding within the bulk, which empirically results in lower surface energy of the corresponding facet along denser packing direction. Consequently, the facet perpendicular to these directional intermolecular chains is expected to have a higher overall surface energy and, therefore, a faster growth rate (Fig. 3c).

In general, the direction with the smaller in-plane lattice parameter corresponds to the longer axis of the crystal (Table 1). However, the exceptions are observed in $(ABA)_2PbI_4$ and $(2CF_3PEA)_2PbI_4$ which exhibit directional hydrogen bonding and π⋯π stacking interactions, respectively. Despite this directional interaction, the in-plane axis with stronger interactions is slightly longer than the other axis. Nevertheless, the crystal growth direction aligns with the directional intermolecular interactions, further emphasizing the potentially more critical role of these interactions in guiding the 1D growth. Furthermore, we have investigated the crystal growth behaviors of several 2D perovskites, including $(ABA)_2PbI_4$, $(PMA)_2PbI_4$, $(2FPMA)_2PbI_4$, $(MBA)_2PbI_4$, $(DFP)_2PbI_4$, and $(DFPD)_2PbI_4$, in non-aqueous solvents such as acetonitrile and tetrahydrofuran (Supplementary Fig. 14). The results show the persistence of 1D crystal growth, highlighting the robustness of the directional intermolecular interaction mechanism in governing the crystal growth.

## Growth of two-dimensional perovskite nanowires

Having understood the mechanism underlying anisotropic crystal growth, we deliberately modified the growth conditions to increase the aspect ratios of these ribbon-like crystals, leading to the formation

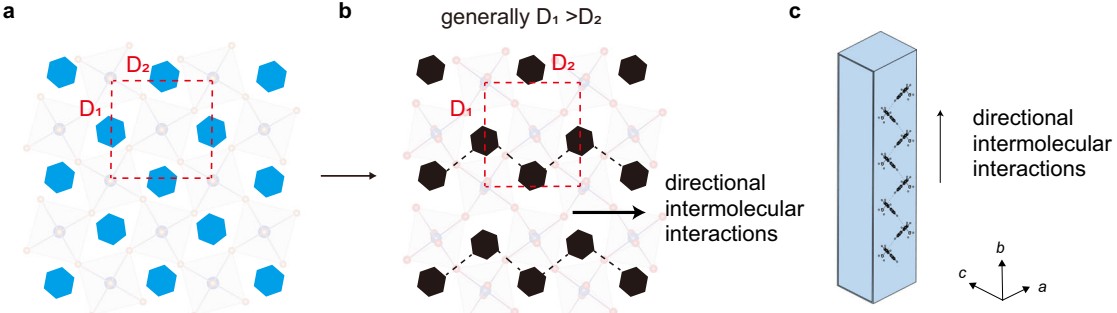

**Fig. 3 | Anisotropic crystal structures and growth induced by directional intermolecular interactions. a**, **b** Schematic illustration of isotropic inorganic lattices (**a**) and anisotropic inorganic lattices (**b**) and their correlation with the absence and presence of directional intermolecular interactions, respectively. **c** Schematic illustration of the needle-like crystal, showing the growth direction aligned with the directional intermolecular interactions.

**Table 1 | Summary of in-plane lattice parameters, types and directions of intermolecular interaction chains, and crystal growth directions for various 2D perovskites**

| Perovskite | Longer-axis $D_1$ (Å) | Shorter-axis $D_2$ (Å) | Anisotropic ratio $D_1/D_2$ | Intermolecular interaction chain | 1D growth direction |
|---|---|---|---|---|---|
| $(PMA)_2PbI_4$ | 9.156 | 8.689 | 1.054 | C–H···π along $D_2$ | along $D_2$ |
| $(2FPMA)_2PbI_4$ | 9.155 | 8.699 | 1.052 | C–H···π along $D_2$ | along $D_2$ |
| $(4FPMA)_2PbI_4$ | 9.244 | 8.696 | 1.063 | C–H···π along $D_2$ | along $D_2$ |
| $(MBA)_2PbI_4$ | 9.311 | 8.894 | 1.047 | C–H···π along $D_2$ | along $D_2$ |
| $(DFP)_2PbI_4$ | 9.289 | 8.982 | 1.034 | Hydrogen bond along $D_2$ | along $D_2$ |
| $(DFPD)_2PbI_4$ | 9.346 | 9.011 | 1.037 | Hydrogen bond along $D_2$ | along $D_2$ |
| $(PA)_2PbI_4$ | 8.930 | 8.672 | 1.030 | $CH_3$···$CH_3$ along $D_2$ | along $D_2$ |
| $(ABA)_2PbI_4$ | 9.280 | 8.906 | 1.042 | Hydrogen bond along $D_1$ | along $D_1$ |
| $(2CF_3PEA)_2PbI_4$ | 8.665 | 8.534 | 1.015 | π···π along $D_1$ | along $D_1$ |
| $(PMA)_2SnI_4$ | 9.094 | 8.667 | 1.049 | C–H···π along $D_2$ | along $D_2$ |
| $(2FPMA)_2SnI_4$ | 9.104 | 8.667 | 1.050 | C–H···π along $D_2$ | along $D_2$ |
| $(PA)_2SnI_4$ | 8.975 | 8.579 | 1.046 | $CH_3$···$CH_3$ along $D_2$ | along $D_2$ |
| $(DFP)_2SnI_4$ | 9.311 | 8.950 | 1.040 | Hydrogen bonds along $D_2$ | along $D_2$ |
| $(DFPD)_2SnI_4$ | 9.501 | 8.872 | 1.071 | Hydrogen bonds along $D_2$ | along $D_2$ |
| $(MBA)_2SnI_4$ | 9.357 | 8.910 | 1.050 | C–H···π along $D_2$ | along $D_2$ |
| $(2CF_3PEA)_2CuCl_4$ | 8.253 | 7.666 | 1.077 | π···π along $D_2$ | along $D_2$ |
| $(CH_2O_2PEA)_2CdCl_4$ | 7.432 | 7.496 | 0.991 | π···π along $D_2$ | along $D_2$ |
| $(TAP3)_2PbBr_4$ | 9.540 | 7.896 | 1.208 | π···π along $D_2$ and hydrogen bond along $D_1$ | along $D_2$ |
| $(BrCA3)_2PbBr_4$ | 8.949 | 7.760 | 1.153 | π···π along $D_2$ | along $D_2$ |
| $(BA)_2PbI_4$ | 8.876 | 8.693 | 1.021 | Weak $CH_3$···$CH_3$ along $D_2$ | Near 2D growth |
| $(HA)_2PbI_4$ | 8.941 | 8.687 | 1.029 | Weak $CH_3$···$CH_3$ along $D_2$ | Near 2D growth |
| $(PEA)_2PbI_4$ | 8.744 | 8.744 | 1.000 | no directional non-covalent interaction | 2D growth |
| $(4CF_3PMA)_2PbI_4$ | 8.691 | 8.418 | 1.032 | no directional non-covalent interaction | 2D growth |
| $(PMA)_2PbBr_4$ | 8.147 | 8.123 | 1.003 | no directional non-covalent interaction | 2D growth |

of single-crystal NWs with high aspect ratios (Fig. 4a). Previous research have shown the importance of introducing dislocation sources and maintaining a low supersaturation to facilitate 1D NW growth[20,26]. We first choose synthesis conditions (see details on precursor concentrations and synthesis conditions in Supplementary information) that maintain a low supersaturation condition. We further employed methods such as scratching the substrate or agitating the precursor solution with a pipette tip, to induce enough seeding particles. The 2D perovskites with isotropic growth habits predominantly formed 2D thin sheets, whereas those with anisotropic growth habits yielded NWs. Figure 4b, c show optical images of the as-grown $(PMA)_2PbI_4$ NWs grown without and with manually introducing nucleation, and the growth process of $(PMA)_2PbI_4$ NWs is available in Supplementary Movie 1. These as-grown NWs in solution can be readily transferred from the solution using PDMS film for structural and optical characterization (Supplementary Fig. 15). We characterized the morphology of the representative $(PMA)_2PbI_4$ NWs using scanning electron microscope (SEM and inset in Fig. 4c) and AFM (Fig. 4d), showing that the NW has a rectangular cross-section with smooth facet, forming high-quality Fabry-Perot cavities. The distribution of the width and length of 80 $(PMA)_2PbI_4$ NWs is illustrated in Fig. 4e, exhibiting an aspect ratio up to 200. Note that the thin NWs were easy to break during transfer, leading to underestimated aspect ratios. AFM measurement on 12 selected NWs revealed $204 \pm 85$ nm in thickness, which is about half of the width ($404 \pm 110$ nm, Fig. 4f). Control experiments without introducing nucleation yielded microrods with smaller aspect ratios (Fig. 4e). Powder XRD pattern of the NWs exhibits predominant diffraction peaks corresponding to the (001) planes, suggesting these NWs lying on these faces (Supplementary Fig. 15). We also attempted to perform electron diffraction measurement on a single NW, but the samples were very unstable under exposure to the electron beam.

We show that many 2D perovskites that form elongated needle-like or ribbon-like morphologies (Fig. 1d) can be engineered to produce NWs, whereas those forming plate-like morphologies cannot (Fig. 1c). We have successfully synthesized NWs of a wide range of 2D perovskites (up to 25 phases) with varying spacer cations, metal cations, halide anions, cage cations, and inorganic layer thickness ($n$ value) (Fig. 4g and Supplementary Fig. 16 for SEM images). The resulting products were confirmed in the 2D perovskite phases by powder X-ray diffraction measurements and PL measurement (Supplementary Fig. 17). For spacer cations such as $ABA^+$ and $MBA^+$ which exhibit very strong tendency of forming needles, NWs with high aspect ratios can also be formed at low supersaturation without introducing nucleation. The statistical distributions of the lateral dimensions for other 2D perovskites, such as $(MBA)_2PbI_4$, $(ABA)_2PbI_4$, and $(DFPD)_2PbI_4$ NWs, are provided in Supplementary Fig. 18.

It is worth noting that spacer cations capable of forming 2D lead iodide perovskite NWs typically can also form 2D tin iodide perovskite NWs (Supplementary Fig. 19) due to the preservation of the directional intermolecular interactions. 2D lead bromide perovskite NWs can also be obtained with spacer cations such as $2BrPEA^+$, while cannot with many others due to the transformation of the anisotropic into isotropic intermolecular interactions in the bromide structures. For $n = 2$ RP perovskites, the incorporation of large cage cations, such as guanidinium ($GA^+$) and acetamidinium ($ATA^+$), enhances the likelihood of forming NWs. For example, $(HA)_2(GA)Pb_2I_7$ forms NWs, whereas $(HA)_2(MA)Pb_2I_7$ tends to form 2D ribbons with smaller aspect ratios. This is because the incorporation of the larger $GA^+$ cation expands the lead iodide framework, causing the alkyl tails of the $HA^+$ cations to displace more away from the center of four neighboring cations, and thereby increasing the anisotropy of intermolecular interactions (Supplementary Fig. 20 and Supplementary Fig. 2).

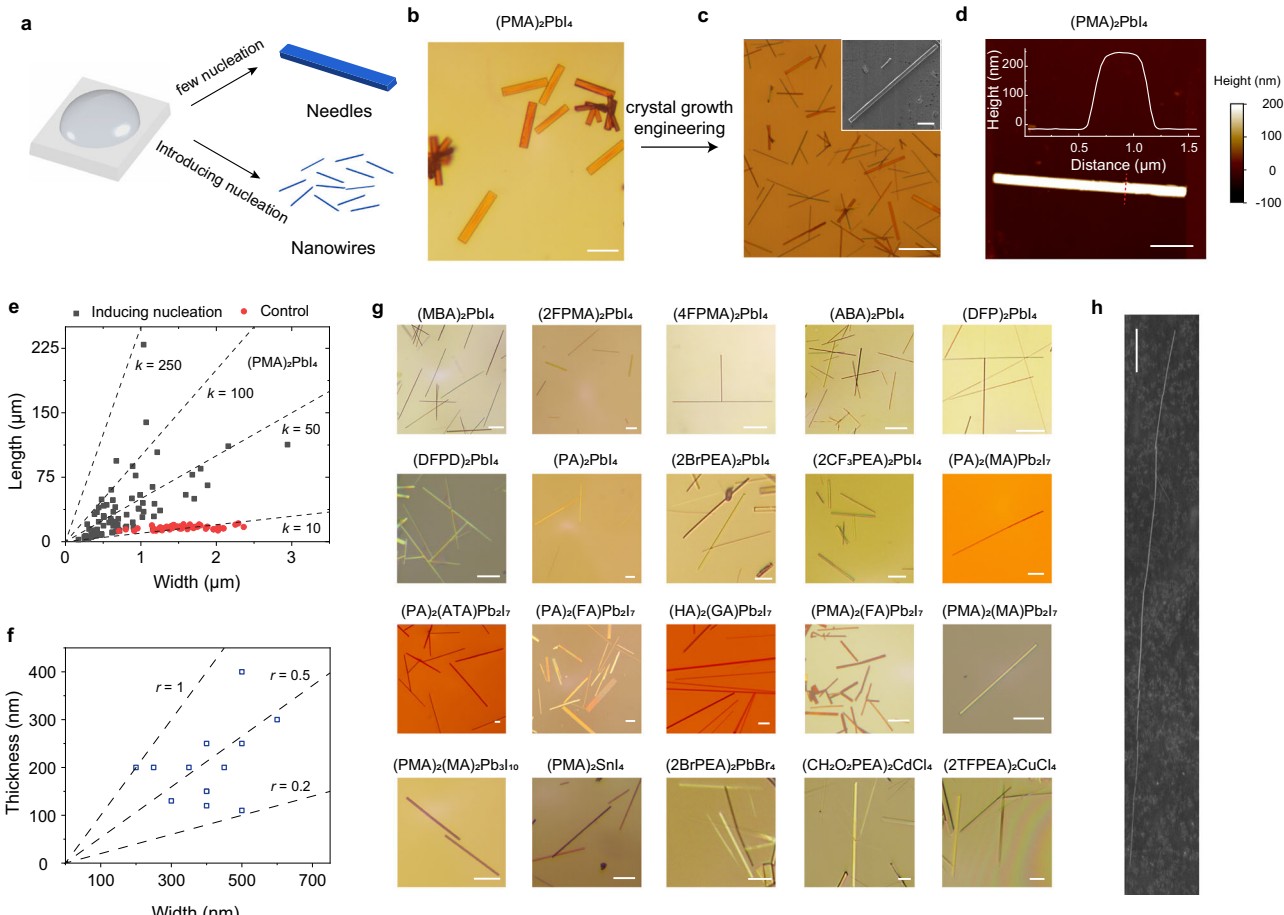

**Fig. 4 | Growth and characterization of 2D perovskite nanowires with different compositions. a** Scheme for the growth of NWs with large aspect ratio by introducing nucleation. **b, c** Comparison of $(PMA)_2PbI_4$ NWs grown with (**c**) and without (**b**) scratching the substrate. Scale bar in B is 10 μm. Scale bar in C is 30 μm. Inset is scanning electron microscope (SEM) image of a $(PMA)_2PbI_4$ NW with flat surface. Scale bar is 5 μm. **d** Atomic force microscope image of a $(PMA)_2PbI_4$ NW. Inset is the height profile extracted along the dashed line. Scale bar is 2 μm. **e** Statistics of the width and length distributions of $(PMA)_2PbI_4$ grown under conditions with and without introducing nucleation. The slope ($k$) indicates the aspect ratio. **f** Statistics of the width and thickness distributions of 12 selected $(PMA)_2PbI_4$ NWs. **g** Optical images of 2D perovskite NWs with various spacer cations, metal cations, halide anions, cage cations, and $n$ values. All scale bars are 10 μm. $CH_2O_2PEA^+$ = 3,4-methylenedioxyphenethylammonium. **h** SEM image of a $(DFPD)_2PbI_4$ NW with a high aspect ratio. Scale bar is 10 μm.

Previous 2D perovskite NWs have been limited to several specialized spacer cations, which tend to exhibit larger structural distortions and less ideal photophysical properties[29]. In contrast, we have NW examples with spacer cations employing derivatives of $PEA^+$ that give rise to excellent photophysical properties among the reported spacer cations[41,42]. This can be readily seen by the smaller octahedral distortion parameters and more symmetric PL emission peak with less peak width in our examples (Supplementary Fig. 21). Moreover, our mechanistic insights have allowed us to not only fingerprint existing 2D perovskites capable of forming NWs based on their crystal structures but also to rationally design and discover structures. We have discovered three 2D tin perovskite structures, including $(DFP)_2SnI_4$, $(DFPD)_2SnI_4$, and $(2FPMA)_2SnI_4$. The generality of our method can be even extended to transition metal cations such as cadmium and copper (Fig. 4g and Supplementary Fig. 22), specifically $(CH_2O_2PEA)_2CdCl_4$ ($CH_2O_2PEA^+$ = 3,4-methylenedioxyphenethylammonium) and $(2CF_3PEA)_2CuCl_4$.

**Robust exciton-polaritons and lasing in nanowires**

Supplementary Fig. 23 shows PL images of various 2D perovskite NWs, demonstrating wide color tunability. These 2D perovskite NWs exhibit anisotropic exciton emission, attributed to in-plane confinement of the exciton. This mechanism differs from conventional inorganic semiconductors, where polarized PL occurs only when the diameter is much smaller than the wavelength due to dielectric contrast. The degree of polarization of representative NWs with $n$ = 1, 2, and 3 is 78%, 39%, and 46%, respectively (Fig. 5a and Supplementary Fig. 24). Compared to 3D $MAPbI_3$ NWs, 2D perovskite NWs can exhibit comparable air, thermal, and photoluminescence stability, which vary depending on their compositions (Supplementary Fig. 25)[29,43].

Strong exciton-photon coupling results in the formation of exciton-polaritons, manifested by an avoided crossing of the photon and exciton dispersion curves. The robustness of polaritons is determined by both the exciton binding energy and the Rabi splitting energy ($\Omega$), with the latter measuring the coupling strength and being proportional to the square root of the ratio of the exciton oscillator strength ($f$) to the mode volume ($V$), i.e., $\Omega \propto \sqrt{f/V}$. Previous efforts to increase the Rabi splitting energies have been focused on engineering the NW dimensionality[44–46], these available 2D perovskite NWs with tunable quantum-well structures provide a paradigm for exploring polaritons.

We investigated the exciton-photon coupling in these single-crystal NWs. The $(PMA)_2PbI_4$ NWs exhibit a narrow PL peak at 2.319 eV with a full width at half-maximum (FWHM) of 130 meV (Supplementary Fig. 26). The exciton binding energy is about 520 meV, which surpasses that of 3D $CsPbBr_3$ by an order of magnitude due to quantum and

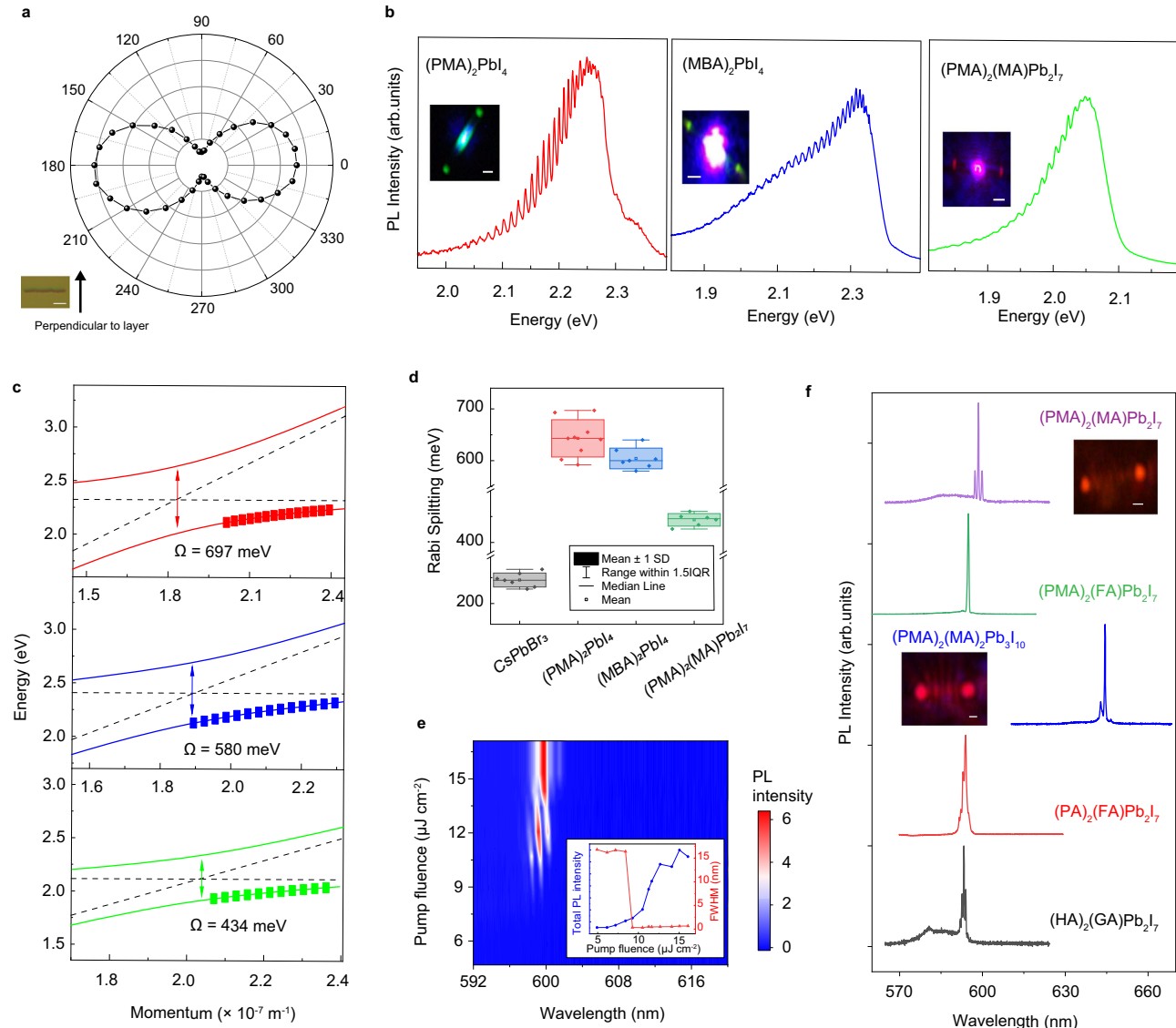

**Fig. 5 | Robust exciton-photon coupling and lasing in 2D perovskite nanowires.** **a** Polarization dependent PL emission of a $(DFPD)_2PbI_4$ showing a degree of polarization up to 80%. Inset is the optical image of the microwire, indicating the alignment. Scale bar is 10 μm. **b** Spatially resolved photoluminescence (PL) spectra collected from the end of three NWs: $(PMA)_2PbI_4$, $(MBA)_2PbI_4$, and $(PMA)_2(MA)Pb_2I_7$. Scale bars are 2 μm. **c** Dispersion curves for the three NWs. The dots are the experimental data, the solid lines are the fitting dispersion curves of the polaritons, and dashed lines are the dispersion curves of the photons or excitons. **d** Statistics of the Rabi splitting energy of 3D perovskite NWs and various 2D perovskite quantum-well NWs. SD represents the standard deviation, and IQR represents the interquartile range. **e** 2D color plot of the emission spectra of a $(PMA)_2(MA)Pb_2I_7$ NW under different pump fluences. Inset shows the plots of integrated PL emission intensity and FWHM of the emission peaks versus the pump fluences. **f** Lasing emission spectra of various NWs of 2D perovskites with $n > 1$. Insets are the optical images of the NWs above the lasing thresholds, showing interference. Scale bars are 1 μm.

dielectric confinement[47]. Using spatially resolved PL spectroscopy, we collected the emission spectrum from the end of the NW (Fig. 5b). An optical image of the NW under laser excitation (inset in Fig. 5b) shows bright emission spots at both ends of the NWs, a result of waveguiding effect. The waveguided emission is significantly red-shifted relative to local emission due to band tail state induced self-absorption. Additionally, the guided emission spectrum exhibits a series of oscillation peaks. By examining a series of NWs with varied length, we analyzed the mode spacing, i.e., energy difference between two adjacent oscillation peaks, at a same wavelength in relation to the NW length. The results reveal a linear relationship, confirming the Fabry-Perot cavity modes in the NWs (Supplementary Fig. 27). However, for a specific NW, the mode spacings are not uniform but decrease as the mode energy increases (Fig. 5c), deviating from the behavior of a pure photonic mode. This irregular mode spacing suggests the formation of polaritons[48,49].

To quantify the strength of exciton-photon coupling, we derived the energy dispersion using the dielectric function model as previously described (see Methods for more details). Figure 5c shows the dispersion relationship of the polaritons, where squared dots represent experimental data, and solid lines the modeled dispersions, revealing a $\Omega$ of 697 meV for this $(PMA)_2PbI_4$ NW. Several factors, including crystalline quality and cross-sectional dimensions, can influence the $\Omega$ in these NWs. Measurements from 9 individual NWs revealed $\Omega$ ranging from 600 to 700 meV, averaging at $640 \pm 30$ meV (Fig. 5d and Supplementary Fig. 28). These values are on par or even exceed those of organic semiconductors[50,51], known for their strong exciton-photon coupling. We also examined the exciton-photon coupling in 3D $CsPbBr_3$ NWs (Supplementary Fig. 29), revealing an average $\Omega$ of $250 \pm 20$ meV, consistent with previous studies[45,52]. The significantly higher $\Omega$ in these NWs can be mainly attributed to their larger exciton

binding energy. Exciton-photon coupling in other NWs with varying spacer cations and quantum-well thickness, such as $(MBA)_2PbI_4$ and $(PMA)_2(MA)Pb_2I_7$, was also investigated (Fig. 5b, c). The $(MBA)_2PbI_4$ NWs exhibit a relatively broad emission peak at 2.405 eV with a FWHM of 146 meV (Supplementary Fig. 26). The average $\Omega$ was $590 \pm 30$ meV, which is smaller than that observed for $(PMA)_2PbI_4$ NWs, attributed to a faster exciton dissipation rate, as indicated by the larger FWHM. $(PMA)_2(MA)Pb_2I_7$ NWs display a narrow PL peak at 2.114 eV with a FWHM of 100 meV (Supplementary Fig. 26). The average $\Omega$ was $440 \pm 20$ meV, slightly lower than that of $(PMA)_2PbI_4$ NWs, attributed to a smaller exciton binding energy due to thicker inorganic layer.

We measured the waveguide propagation loss coefficient of the NWs to be 4 dB mm[-1], which are comparable to the best-performance inorganic and organic semiconductor waveguides[1,53,54] (Supplementary Fig. 30). Consequently, many of these well-faceted high-quality NWs are capable of lasing under femtosecond pulsed laser excitation. Figure 5e shows the 2D color plot of the PL spectra of a $(PMA)_2(MA)Pb_2I_7$ NW with increasing pump fluence from 5 to 16 µJ cm[-2] at 80 K. The inset depicts the integrated PL intensity and the FWHM of the PL peak as functions of the pump fluence. Below the lasing threshold of 9.2 µJ cm[-2], the NW is characterized by a broad PL peak centered at 590 nm with a FWHM of ~17 nm, alongside a gradual increase in the integrated PL intensity with increasing pump fluence. Above the threshold, several distinct and sharp lasing peaks become evident, showing a rapid rise in intensity as the pump fluence increases, whereas the intensity of the spontaneous emission region nears saturation (Supplementary Fig. 31), indicating the onset of lasing. The FWHM of the primary lasing peak, located at 600 nm, is 0.4 nm, corresponding to a quality factor of ~1500. We found that lasing can occur in 2D lead iodide perovskite NWs with $n > 1$ (Fig. 5f), regardless of the choices of spacer cations or cage cations, but not in those 2D lead iodide perovskites with $n = 1$. Additional lasing studies of other NWs are provided in Supplementary Fig. 31 and Supplementary Fig. 32. Previous studies underscore the importance of specially designed π-conjugated spacer cations for enabling lasing from the $n = 2$ structure[12,55]. However, we observed efficient lasing from 2D RP perovskites incorporating aliphatic cations such as $(PA)_2(FA)Pb_2I_7$ and $(HA)_2(GA)Pb_2I_7$, with lasing thresholds comparable to that of $(PMA)_2(MA)Pb_2I_7$, attributable to high-quality cavities of our NWs. Moreover, compared to the exfoliated single-crystal samples, the NWs of the same perovskite exhibit significantly lower lasing thresholds (i.e., ~4 times lower) due to their well-defined 1D geometry (Supplementary Fig. 33).

## Discussion

The ability to control 1D crystal growth of semiconductors has led to significant advancements in nanoscience and nanotechnology. 2D organic-inorganic hybrid semiconductors, a current research frontier, offer promising properties in optoelectronics, spintronics, ferroelectrics, and layered ferromagnetism. Our work demonstrates a fundamentally crystal growth mechanism that utilizes directional noncovalent intermolecular interactions to synthesize 1D forms of diverse 2D metal halide perovskites. Enhanced intermolecular interactions among the spacer cations along a certain basal plane direction in selected structures induce the preferential anisotropic growth of crystals. This in combination with rationally controlled crystal growth enable the solution synthesis of 1D NWs of diverse 2D perovskites with growth axes aligned parallel to the in-plane direction, marking a general approach for NW growth of hybrid materials.

Through our work, we can predict the likelihood of a given structure forming NWs based on its crystal structure using our synthetic approach. This capability, in part, explains our success in establishing a broad library of 2D perovskite NWs with diverse spacer cations. However, predicting whether a specific spacer cation would induce directional intermolecular interactions in 2D perovskite structures—or generalizing its molecular motif—remains challenging. This is

most directly reflected by the observation that even when using the same spacer cation, such as PMA[+], the bromide-based structure exhibits different intermolecular interactions compared to its iodide counterpart. Future research on the rational molecular design of spacer cations to promote such interactions is highly desirable and will require further theoretical and experimental investigations.

Optical studies have revealed significantly enhanced exciton-photon coupling in these quantum-well NWs compared to 3D semiconductors, as well as wavelength-tunable and more efficient lasing relative to exfoliated crystals of 2D perovskites. Furthermore, these NWs demonstrate strong anisotropic exciton emission due to the in-plane confinement of the exciton. By further embedding the NWs between two dielectric mirrors along the in-plane direction, usual anisotropic polaritons may be realized, warranting future investigation[13,56]. Additionally, the NW growth can be extended to a broad range of 2D metal halide perovskites, which exhibit nonlinear optical effects, chirality, and ferroelectric and ferromagnetic properties[57,58]. These attributes, combined with large Rabi splitting and low-loss waveguide, make these NWs as compelling candidates to achieve high-speed light modulation and processing in nanowire-based photonic devices and circuits.

## Methods
### Starting Materials
$PbI_2$, CsBr, $PbBr_2$, $Sn(Ac)_2$, HI (47% wt% in $H_2O$, stabilized with 1.5% $H_3PO_2$), HBr (48% wt% in $H_2O$), $H_3PO_2$(50 wt% in $H_2O$), dimethyl sulfoxide, phenylmethylamine (PMA), phenylethylamine (PEA), 4-fluorophenylmethylamine (4FPMA), 2-fluorophenylmethylamine (2FPMA), 4-chlorophenylmethylamine (4ClPMA), 4-trifluoromethylphenylmethylamine ($4CF_3PMA$), n-butylamine (BA), 4-aminobutyric acid (ABA), n-amylamine (PA), n-hexylamine (HA), 4,4-difluoropiperidine (DFPD), 3,3-difluoropyrrole hydrochloride, R-methylbenzylamine (MBA), 2-trifluoromethylphenylethylaine ($2CF_3PEA$), 2-bromophenylethylamine (2BrPEA), 3,4-methylenedioxyphenethylamine ($CH_2O_2PEA$), methylamine hydrochloride, formamidine hydrochloride, guanidine hydrochloride, and acetamidine hydrochloride were purchased from commercial suppliers, including Sigma-Aldrich, and Macklin. All the chemicals were used as received without further purification.

### Growth of two-dimensional perovskite crystals
The precursor solutions of $(LA)_2PbI_4$ were prepared by adding 1 mmol of $PbI_2$ and 2 mmol of the organic amine in a mixed solution containing 10 mL of HI and 1 mL of $H_3PO_2$ in a vial. Given the significantly high solubility of $(ABA)_2PbI_4$, the volume of the acidic solution was deceased to 2.5 mL. The precursor solution of $(PMA)_2(MA)Pb_2I_7$ was prepared by adding 1.5 mmol of $PbI_2$, 1 mmol of benzylamine, and 8 mmol of methylamine hydrochloride in a mixed solution containing 6 mL of HI and 1 mL of $H_3PO_2$. The precursor solutions of $(PA)_2(MA)Pb_2I_7$ were prepared by adding 1 mmol of $PbI_2$, 1 mmol of n-amylamine, and 2 mmol of methylamine hydrochloride in a mixed solution containing 5 mL of HI and 1 mL of $H_3PO_2$. The precursor solution of $(PA)_2(FA)Pb_2I_7$ was prepared by adding 1 mmol of $PbI_2$, 1 mmol of $n$-amylamine, and 0.6 mmol of formamidine hydrochloride in a mixed solution containing 4.5 mL of HI and 0.5 mL of $H_3PO_2$. The precursor solution of $(PA)_2(ATA)Pb_2I_7$ was prepared by adding 2 mmol of $PbI_2$, 1.2 mmol of $n$-amylamine, and 1 mmol of methylamine hydrochloride in a mixed solution containing 6 mL of HI and 0.5 mL of $H_3PO_2$. The precursor solution of $(HA)_2(MA)Pb_2I_7$ was prepared by adding 2 mmol of $PbI_2$, 0.9 mmol of $n$-hexylamine, and 1.0 mmol of methylamine hydrochloride in a mixed solution containing 2.5 mL of HI and 0.3 mL of $H_3PO_2$. The precursor solution of $(HA)_2(GA)Pb_2I_7$ was prepared by adding 2 mmol of $PbI_2$, 0.5 mmol of $n$-hexylamine, and 1 mmol of guanidine hydrochloride in a mixed solution containing 3 mL of HI and 0.3 mL of $H_3PO_2$. The precursor solution of $(PMA)_2(MA)_2Pb_3I_{10}$ was

prepared by adding 2.5 mmol of $PbI_2$, 1 mmol of benzylamine, and 9 mmol of methylamine hydrochloride in a mixed solution containing 5 mL of HI and 1 mL of $H_3PO_2$. The precursor solution of $(PMA)_2PbBr_4$ was by adding 1 mmol of $PbBr_2$ and 2 mmol of benzylamine in a mixed solution containing 10 mL of HBr and 1 mL of $H_3PO_2$ in a vial. The precursor solutions of $(LA)_2SnI_4$ were prepared by adding 1 mmol of tin (II) acetate and 2 mmol of organic amine into a mixed solution containing 6 mL of HI and 2 mL of $H_3PO_2$ in a vial. The recipes for all perovskite precursor solutions are summarized in Table S1. The precursor solution of $(2CF_3PEA)_2CuCl_4$ was prepared by adding 0.5 mmol of $CuCl_2$ and 1 mmol of 2-trifluoromethylphenylethylaine in a mixed solution containing 0.2 mL of HCl and 3 mL of $H_2O$ in a vial. The precursor solutions of $(CH_2O_2PEA)_2CdCl_4$ were prepared by adding 0.5 mmol of $CdCO_3$ and 1 mmol of 3,4-methylenediox-yphenethylamine in a mixed solution containing 0.2 mL of HCl and 3 mL of $H_2O$ in a vial.

These precursor solutions were heated and stirred on a hot plate at 110 °C until the complete dissolution of the solids. Subsequently, the heated solutions were cooled to near room temperature (~20 °C), leading to precipitation of 2D perovskite crystals. The growth of various microstructures of 2D perovskites was observable under optical microscope upon dispensing a droplet of the clear solution onto a glass slide or ITO substrate. The as-grown microstructures were transferred from the solution by contacting the crystals with a poly-dimethylsiloxane film (PDMS, from GelPak, PF-30-X4) for morphology and optical characterization. For the growth of large single crystals, the precursor solutions were prepared as described above. After the precipitation of the 2D perovskite crystals, the top clear and saturated solutions were transferred into a sperate vial, to which a few pieces of 2D perovskite crystals were added. The obtained solutions were heated at 110 °C to dissolve the crystals, and then left undisturbed at room temperature. Large single crystals formed at the bottom of solution within a few days. For the growth of plate-like $(PA)PbI_4$ crystals, the precursor solution was placed on the hot plate at 340 K, and large crystals formed within a few hours.

## Synthesis of perovskite nanowires

The 2D perovskite NWs were synthesized under a low supersaturation. The precursor solutions of $(LA)_2PbI_4$, $(LA)_2SnI_4$, and $(PMA)_2(MA)_{n-1}PbI_{3n+1}$ ($n = 2$ and 3) were initially warmed to about 40 °C, followed by dispensing of 10 μL onto a glass slide or an ITO substrate at room temperature using a pipette. Scratching the glass slides rapidly with the pipette tip introduces enough nucleation, leading to the formation of NWs with large aspect ratios. For spacer cations such as $ABA^+$ and $MBA^+$ which exhibit very strong tendency of forming needles, NWs with high aspect ratios could also be formed at low supersaturation without manually introducing nucleation. The NWs were picked-up by a polydimethylsiloxane film (PDMS, from GelPak, PF-30-X4) and could be transferred onto Si/SiO$_2$ substrate for optical characterization through pressing the PDMS film on the substrates. To inhibit their growth in thickness and width, the NWs were promptly separated from the solution within minutes. To synthesize tin-based NWs, the dispensed solutions were sandwiched between two slides to prevent oxidation after scratching the substrate. For the growth of $(LA)_2(A)Pb_2I_7$ NWs ($LA^+ = PA^+$ or $HA^+$, $A^+ = MA^+$, $FA^+$, $GA^+$, or $ATA^+$), the cold precursor solutions (10 μL) were dispensed onto a glass slide, and the substrates were scratched when the crystals started to precipitate. The growth time varied from a few to tens of minutes.

The growth of 2D perovskite NWs in non-aqueous solvents was achieved by solvent evaporation at room temperature. Precursor solutions were prepared by dissolving 50 mg of various 2D perovskite crystals, including $(ABA)_2PbI_4$, $(PMA)_2PbI_4$, $(2FPMA)_2PbI_4$, $(MBA)_2PbI_4$, $(DFP)_2PbI_4$, and $(DFPD)_2PbI_4$, in 200 μL of acetonitrile or tetra-hydrofuran. The resulting solution was then placed between the slits of two glass slides to control the evaporation rate.

$CsPbBr_3$ nanowires were synthesized on mica substrates by chemical vapor deposition. A $CsPbBr_3$ precursor solution was prepared by dissolving 5.6 mmol of CsBr and 5.6 mmol of $PbBr_2$ in 10 mL of dimethyl sulfoxide. A droplet of the solution was then introduced onto a glass slide that has been preheated to ~300 °C. The solvent rapidly evaporated, leaving behind $CsPbBr_3$ solid. A piece of exfoliated mica substrate was placed onto the $CsPbBr_3$, and NWs were formed on the mica substrate within 2 min.

## Structural characterizations

The optical images of micro/nanocrystals were obtained using an optical microscope (AOSVI). The atomic force microscope (AFM) was performed on a Bioscrope AFM using ScanAsyst-Air tips from Bruker AFM Probes (spring constant k: 40 N m$^{-1}$). The various samples for AFM were prepared by crystallizing at low supersaturation and then transferred onto Si/SiO$_2$ substrates. The single-crystal X-ray diffraction measurements were performed using an XtaLAB PRO 007HF (Mo) single-crystal X-ray diffractometer with Mo Kα radiation ($\lambda = 0.71073$ Å). The crystals for SCXRD were placed in a specific orientation and the corresponding X-ray diffraction images were collected. The powder X-ray diffraction data were collected using an X-Pert3 powder X-ray diffractometer with Cu Kα radiation ($\lambda = 1.54056$ Å). The NW samples were transferred from the solution and placed on the PDMS film. The scanning electron microscopy images were collected using a field-emission scanning electron microscope (S-4800, Hitachi) with an accelerating voltage of 2 kV. The samples were transferred onto Si/SiO$_2$ substrates and dried overnight at 40 °C.

## Optical characterizations

The PL and absorption spectra of various samples were acquired on a homebuilt microscope system based on Olympus IX73 inverted microscope. A 405 nm continuous-wave laser (TEM-F-405 nm) as the excitation source was focused onto the sample by an objective (40×, NA = 0.65, Olympus) for the PL measurement. Through a par of achromatic lens, the samples were imaged through the entrance slit of the spectrometer (Princeton Instruments, HRS-300S) onto a liquid-nitrogen-cooled 2D charge-coupled device camera (Princeton Instruments, PyLoN-400BRX), as shown in Supplementary Fig. 34. The position of the entrance slit allows for spatial collection of the emission spectra from the local excitation spot or the two ends of the NW. For polarization-resolved PL measurements, a linear polarizer (LPVISE100-A) was added before the entrance of microscope to obtain linear polarized light, and another linear polarizer and a half-wave plate (THORLABS, AHWP10M-580) were placed before the entrance slit to analyze the emission polarization. The absorption spectra of the samples were measured in a transmission geometry with a stabilized tungsten light source (Thorlabs, SLS201L(/M)). The white light was focused onto the sample by a long working distance objective (40×, NA = 0.65, Nikon), and the transmission light were collected by another objective (40×, NA = 0.65, Nikon) and detected by a fiber optic spectrometer. Lasing measurements under low temperature were conducted with a cryostat (Scryo-S-300MS). The excitation pulses (centered wavelength at 400 nm, 200 Hz) were generated from the optical parametric amplifier (ORPHEUS-HP), which was seeded by the femtosecond lasers (PHAROS-PH2-20W, 1030 nm, 290 fs, 200 kHz). The excitation laser was expanded by a len before entering the microscope. The power was adjusted by a neutral density filter. The emissions from the nanowires were collected by the same objective and detected by the spectrometer as mentioned above. The grating is 1200 mm$^{-1}$. Wide-field photoluminescence photos were captured using the imaging system of Olympus IX73 inverted microscope. The nanowires were excited by a mercury lamp, and the emissions were detected by a color camera (SONY, E31SPM25000KPA), after passing through a 425 nm long-pass emission filter (THORLABS, DMLP425).

The waveguide loss measurements were carried out on a Q2 laser scanning confocal microscope (ISS). The excitation laser was focused onto the sample by a 20× objective (Nikon), and the emission images was collected on a CCD camera after passing a 480 nm long-pass filter, the excited spot was adjusted by the scanning galvanometer. The waveguide loss coefficient was estimated by the following formula, $\frac{I_{tip}}{I_{body}} = Ae^{-\alpha D}$. In this model, $I_{tip}$ is the fluorescence intensity of the end of the nanowires, $I_{body}$ is the fluorescence intensity of the excitation spot, $\alpha$ is the waveguide loss coefficient, $D$ is the distance between the excitation spot and the end.

**Dispersion relation for exciton polaritons**

The energy dispersions of the exciton-polaritons were derived using the dielectric function model, $E(\omega, k) = \hbar\omega(k) = \frac{\hbar c k}{\sqrt{\epsilon_b \left(1 + \frac{E_L^2 - E_T^2}{E_T^2 - iE\hbar\gamma - E^2}\right)}}$. In this

model, $\hbar$ is the reduced Plank constant, $k$ the wave vector, $c$ the velocity of light in a vacuum $\epsilon_b$ the background dielectric constant, $E_L$ the longitudinal exciton energy (i.e., the PL peak energy) $E_T$ the transverse exciton energy, $E$ the mode energy, and $\gamma$ the damping constant (i.e., the PL peak's fwhm). For the confined modes in a NW, the wave vector is given by $k = k_0 + m\frac{\pi}{L}$ with $m$ (equals 0, 1, 2, 3…) being mode index of the oscillation peaks, $L$ the length of the NW, and $k_0$ a fitting parameter accounting for the absence of knowledge on the wavevector of the lowest energy mode. When the cavity photon and exciton dispersions intersect at a momentum away from zero, Rabi splitting results in an avoided crossing and the formation of a lower polariton branch and an upper one. The lower polariton branch exhibit a negative curvature, accounting for the nonlinear dispersion. The Rabi splitting energy can be estimated as $\Omega \approx \sqrt{2E_T(E_L - E_T)}$.

**Reporting summary**

Further information on research design is available in the Nature Portfolio Reporting Summary linked to this article.

## Data availability

Crystallographic data for the structures reported in this Article have been deposited at the Cambridge Crystallographic Data Centre, under deposition numbers CCDC: $(PMA)_2SnI_4$-2375615; $(PA)_2SnI_4$-2375614; $(CH_2O_2PEA)_2CdCl_4$-2375613; $(DFPD)_2SnI_4$-2375612; $(DFP)_2SnI_4$-2375611 ; $(2FPMA)_2SnI_4$-2375610 ; $(2CF_3PEA)_2CuCl_4$-2375609. These data can be obtained free of charge from The Cambridge Crystallographic Data Centre via www.ccdc.cam.ac.uk/data_request/cif. Checkcif files for the perovskite crystals are also provided in Supplementary Data 1. The data that support the findings of this study are available from the corresponding author upon request. Source data are provided with this paper.

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

## Acknowledgements
This work is supported by the National Natural Science Foundation of China (22271006), Peking University Instrument Development and Key Technology Research and Development Fund, and the Fundamental Research Funds for the Central Universities, Peking University. Dr. Guan thanks support from Research and Development Program, Lianyungang Center of Institute for Molecular Engineering, Peking University.

## Author contributions
Y.F., M.Z., and L.J. initiated the project. M.Z. synthesized the samples and performed the structual characterizations with the help of L.J., T. Z., and X.J.. M.Z. performed the optical characterizations with the help of L.J., Y. G., and M.L., Y.F. supervised the project. Y.F. and M.Z. wrote the manuscript. All authors have interpreted the findings, commented on the paper, and approved the final version.

## Competing interests
The authors declare no competing interests.
