## [Transparent Peer Review file · Nature Communications]

Two-dimensional Organic-Inorganic Hybrid Perovskite Quantum-Well Nanowires Enabled by Directional Noncovalent Intermolecular Interactions

Corresponding Author: Dr Yongping Fu

Version 0:

Reviewer comments:

Reviewer #1

(Remarks to the Author)

In this manuscript, Zhang, Jin, Fu, and coauthors presented a novel crystal growth mechanism that enabled controllable 1D growth of layered halide perovskites, which exhibited significant Rabi splitting energy and efficient optically pumped lasing. The growth mechanism was generally well explained without major technical issues, and the reported photonic properties were clear and promising. The reviewer acknowledges the authors' extensive efforts and is inclined to support the work for publication. However, the authors should first address the following questions:

1. What is the purpose of showing the reciprocal lattice reconstruction in e.g. Fig 2b or Fig S3, d-e? Were they simulated from solved crystal structures or were they experimentally measured by e.g. electron microscopy? They did not seem to help with the authors' explanations.

2. The authors should try to further elucidate why transitioning from lead iodide or tin iodide framework to lead bromide would attenuate the anisotropic non-covalent interaction between spacer cations. This is connected with the explanation on page 10 regarding the structural expansion effect of GA cations in comparison to smaller MA. On page 8, the authors also pointed out that "the directional intermolecular interactions lead to a denser packing of the spacer cations and a more compact arrangement of the lead iodide framework".

While the authors might have pointed to the right direction, which was the expansion of inorganic framework, whether such expansion was anisotropic (w.r.t. a v.s. b axis) was not discussed yet. In other words, the packing density of inorganic framework might have been affected by the anisotropy within organic, which then led to the anisotropic growth of the assembled hybrid lattice. Such a mechanism can be more convincing than merely relying on the organic intermolecular interactions themselves, as the growth of hybrid layered perovskites requires synergy between organic and inorganic. Thus, such discussions should be accompanied by methods to quantify the anisotropy within inorganic structures, and compare these numbers across different systems. Explanations in this regard would complement the authors' current mechanism.

3. Related to comment #2, the reviewer would also like to challenge the comparison between BA, PA, and HA -based systems with similar alkylammonium backbone. While the anisotropy within spacer non-covalent interactions (a axis v.s. b) was attenuated from PA -based system to BA or HA equivalents, it's hard to justify how much attenuation would lead to disappearance of 1D growth. If CH- π interaction was the sole reason giving rise to 1D growth, then the smaller discrepancies in CH₃...CH₃ distances along the two in-plane directions in BA or HA-based systems might also be non-negligible. Then why they cannot grow into 1D structures, even with a fast nucleation using the droplet method?

4. Thus, the reviewer believes that the authors are presenting a different growth mechanism utilizing non-covalent interactions. Yet, this mechanism did not contradict, but rather complemented the mechanism cited on page 8 using solvation effects from carboxylic acids. Particularly, the linear H-bond chain within (ABA)₂PbI₄ differs fundamentally from the carboxylic dimer synthon. The structural flexibility of carboxylic acid -terminated alkyl chain also differs from rigid benzoic acid. Comparison across two mechanisms should be more carefully addressed, rather than merely pointing out the existence of π - π interaction within the cited systems. Hence, the discussions on page 8 should be revised.

The system here also introduced another merit, as they can be processed in non-aqueous solvents. The authors should

expand more on the growth in non-aqueous systems to generalize the solvent effects.

Additionally, a few minor issues should be corrected:

5. Figs. 4a, S20, S29: images of the wire should come with the polar graph to show their alignment.
6. Fig 17d should be removed due to the unreliable comparison between data obtained from different setups.
7. Typos: S15 figure title; page 11 line 240: should be copper; page 14, line 307: should be Fig 27

Reviewer #2

(Remarks to the Author)

The paper '2D Organic-Inorganic Hybrid Perovskite Quantum-Well Nanowires Enabled by Directional Noncovalent Intermolecular Interactions' by M. Zhang and co-authors, aims to demonstrate a crystal growth mechanism that utilises directional noncovalent intermolecular interactions to synthesize 1D forms of diverse 2D metal halide perovskites. The authors synthesise different types of crystals and try to correlate the intermolecular interactions among the spacer cations with the shape of the millimetre-sized crystals.

The most serious problem is that the explanations provided by the authors are not sufficiently convincing and it is not clear which type of interaction (hydrogen bonds? van der Waals?) can promote 1D growth, there is no trend and not all the experimental findings support their hypotheses.

For example:

- on pages 5 and 6, it is shown that directional C-H \cdots π interactions give 1D crystals ((PMA)₂PbI₄) while isotropic C-H \cdots π interactions give 2D-plate morphology ((PMA)₂PbBr₄). However, this no longer applies in the case of BA⁺, HA⁺ and PA⁺ (p. 7) where the intermolecular interactions are anisotropic for all 3 cations but PA⁺ leads to 1D crystals while HA⁺ and BA⁺ produce 2D crystals, so directional noncovalent intermolecular interactions do not seem to play a fundamental role despite what the authors would like to prove.

- at page, lines 131-134: the authors state that 'the halogen- π interactions in 4FPMA are significantly weaker than in 4CIPMA⁺'; can they explain why, taking into account that F is more electronegative than Cl?

- on page 7, lines 135-138 it is not clear how the intermolecular interactions of 4CF3PMA⁺ and PEA⁺ can direct the crystal growth and why.

Furthermore, the introduction does not cite the relevant literature and does not mention the methods already known for obtaining 2D perovskite microwires. They only cite such literature on page 11 lines 229-235. It would be better to have a comprehensive discussion in the introduction to provide a framework for the topic of the article.

Therefore, I do not believe that the level of novelty and soundness of this work make it suitable for publication in a high-level journal like Nature Communications.

Reviewer #3

(Remarks to the Author)

This work presents a universal method for growing one-dimensional (1D) perovskite single crystals using a variety of organic cation templates. It also relates to, and very similar approach with the recent findings reported in Science 384 (6699), 1000-1006, 2024. The potential impact of 1D growth of halide perovskite is not clear. Although the authors discuss growth mechanisms based on molecular interactions, additional factors, such as the intrinsic crystal structure, may influence the process but were not addressed in this paper.

The authors mention that strong hydrogen bonding and aromatic π - π stacking interactions promote 1D anisotropic growth; however, these effects also impact the dimensionality (1D, 2D, or 0D) of the resulting crystal structure. The authors should report all the single crystal structure (cif) files on the materials presented in this paper.

As the crystals are relatively thin, the authors need to report the luminescence stability of the materials to future validate the potential for light emitter applications.

Minor comment: On page 2, the authors wrote, 'semiconductor nanowires have long fascinated scientist.' This should be revised.

Version 1:

Reviewer comments:

Reviewer #1

(Remarks to the Author)

The authors have done an excellent job addressing the previous comments. In particular, the discussions related to comments #2 and #3 are sophisticated and comprehensive, significantly enhancing the general audience's understanding of the crystal structures and growth dynamics of layered perovskites.

As mentioned previously, the synergy between organic and inorganic components is crucial, and their interdependence often resembles a "chicken-and-egg" scenario. Regarding intermolecular interactions, Wulff construction is typically effective for explaining surface energy anisotropy. However, it becomes somewhat non-intuitive when attempting to generalize it to inorganic lattices.

A few follow-up comments:

1) As the discussions on crystal structures have now become quite comprehensive, it may be worth considering how to better present this information in the main text to enhance readability for a general audience. Additionally, please evaluate if any re-arrangements or adjustments in structure might improve the flow and accessibility of the content.

2) In Fig. S14, a few candidates grew into multiple branches from organic solvents (usually 4 branches). What is the mechanism of branching? For those with a cross shape, would each branch follow the same crystallographic direction or different? (i.e. all along a in MBA_2PbI_4 or along a and b?)

3) Please explain the side peaks in multiple PXRD spectra in Fig. S17.

4) page 18, "Additional lasing studies of other NWs are provided in Fig. S17" – Fig number was not correct. Please check other references as well.

5) "Generally, 2D perovskite NWs demonstrate superior air and thermal stability, whereas the photoluminescence stability of 3D perovskite NWs." This statement seems to contradict with the results shown in Fig. S25. At least from the particles investigated, compared with MAPbI_3 , photostability was generally worse, while air and thermal stability comparable.

Reviewer #2

(Remarks to the Author)

The authors in the revised version have added experimental results and further discussion that enriched the work. However, in my opinion, the claim to demonstrate 'an alternative, universal mechanism for the formation of single-crystal NWs of 2D perovskite' remains unproven.

Specifically, the correlation between intermolecular interaction and crystal shape is still elusive. Even in the revised version, no clear trend or molecular design of the organic cation is presented that could predict the formation of nanowires.

Indeed, in the case of $(\text{BA})_2\text{PbI}_4$, $(\text{PA})_2\text{PbI}_4$ and $(\text{HA})_2\text{PbI}_4$, which exhibit very similar organic chains and $\text{CH}_3\text{---CH}_3$ van der Waals interactions, the synthesis must be conducted at significantly different temperatures (RT vs 67°C) to achieve the assumed trend proposed in the manuscript. This discrepancy suggests that additional factors, possibly complementary to intermolecular interactions, might influence crystal shape, particularly when growth is conducted under identical conditions. For these reasons, I believe the paper is not suitable for publication in Nature Communications.

Reviewer #3

(Remarks to the Author)

The authors answered most of the questions in the response letter, I do not have further questions.

Version 2:

Reviewer comments:

Reviewer #1

(Remarks to the Author)

The authors have addressed all the comments very comprehensively. I appreciate the additional efforts to elucidate the branching mechanism.

Reviewer #2

(Remarks to the Author)

The manuscript has significantly improved compared to the initial submission. However, I believe that framing the article as presenting a "universal mechanism for obtaining single-crystal NWs of 2D perovskites" is not entirely correct. The study does not appear to introduce a new synthetic approach; rather, it involves a screening of various organic cations, many of which lead to unidirectional growth. The findings suggest that this growth behavior is linked to molecular interactions, which vary widely and include numerous exceptions. While the screening and XRD analysis provide valuable insights, I do not find the level of novelty sufficient for publication in Nature Communications.

Response to the Reviewers' Comments

We sincerely thank the Reviewers for their time and effort in reviewing our manuscript. We have made substantial revisions to the manuscript to address their critical comments, which have significantly improved the quality of our work. Below, we reproduce the Reviewers' comments in black font and provide our detailed responses in blue.

Reviewer #1:

In this manuscript, Zhang, Jin, Fu, and coauthors presented a novel crystal growth mechanism that enabled controllable 1D growth of layered halide perovskites, which exhibited significant Rabi splitting energy and efficient optically pumped lasing. The growth mechanism was generally well explained without major technical issues, and the reported photonic properties were clear and promising. The reviewer acknowledges the authors' extensive efforts and is inclined to support the work for publication. However, the authors should first address the following questions:

We thank the Reviewer for the support of our work for publication and constructive suggestions. We have carefully addressed the concerns raised and revised our manuscript accordingly. Below are our detailed responses to each of the comments.

1. What is the purpose of showing the reciprocal lattice reconstruction in e.g. Fig 2b or Fig S3, d-e? Were they simulated from solved crystal structures or were they experimentally measured by e.g. electron microscopy? They did not seem to help with the authors' explanations.

We performed single-crystal X-ray diffraction to determine the indexed facets and growth direction of needle-like crystals. The reciprocal lattice reconstructions presented in original Fig. 2b are from single single-crystal X-ray diffraction, and correspond to two specific unit cell directions. We agree with the Reviewer that presenting this information does not significantly help the explanations. In response, we have revised the original Fig. 2 to present the content in a more concise and focused manner.

2. The authors should try to further elucidate why transitioning from lead iodide or tin iodide framework to lead bromide would attenuate the anisotropic non-covalent interaction between spacer cations. This is connected with the explanation on page 10

regarding the structural expansion effect of GA cations in comparison to smaller MA. On page 8, the authors also pointed out that “the directional intermolecular interactions lead to a denser packing of the spacer cations and a more compact arrangement of the lead iodide framework” .

While the authors might have pointed to the right direction, which was the expansion of inorganic framework, whether such expansion was anisotropic (w.r.t. a v.s. b axis) was not discussed yet. In other words, the packing density of inorganic framework might have been affected by the anisotropy within organic, which then led to the anisotropic growth of the assembled hybrid lattice. Such a mechanism can be more convincing than merely relying on the organic intermolecular interactions themselves, as the growth of hybrid layered perovskites requires synergy between organic and inorganic.

Thus, such discussions should be accompanied by methods to quantify the anisotropy within inorganic structures, and compare these numbers across different systems. Explanations in this regard would complement the authors’ current mechanism.

We appreciate the Reviewer for providing these insightful comments. We have addressed the raised points below detail.

1. Enhanced anisotropic intermolecular interactions by inorganic lattice expansion. In $(\text{PMA})_2\text{PbBr}_4$, the PMA^+ cations exhibit nearly isotropic intermolecular interactions and occupy the center of the pockets formed by four lead bromide octahedra. The distances between neighboring PMA^+ cations are uniform, and each PMA forms weak $\text{C-H}\cdots\pi$ interactions with four adjacent cations (Fig. R1). Upon transitioning to $(\text{PMA})_2\text{PbI}_4$, the packing distance of the PMA^+ cations increase to accommodate the expanded inorganic lattice, as the lead iodide framework provides a larger space for the PMA^+ cations. This expansion causes the PMA^+ cations to displace from the center, forming two strong $\text{C-H}\cdots\pi$ interactions along in-plane D2 direction (Fig. R1). This phenomenon is reminiscent of the phase transitions in 3D perovskites, where an undersized A-site cation (e.g., in CsPbBr_3) induces an octahedral tilt with A-cation off-centering when decreasing temperature (*Acta Crystallogr. Sect. B.* **1972**, 28, 3384; *Prog. Inorg. Chem.* **1999**, 48, 1). A similar comparison can be drawn between cubic MAPbBr_3 perovskite and tetragonal MAPbI_3 perovskite (*J. Am. Chem. Soc.* **2022**, 144, 12247; *Nat. Rev. Chem.* **2021**, 5, 838). This suggests the expansion of the inorganic lattice in $(\text{PMA})_2\text{PbI}_4$ facilitates the off-centering displacement of PMA^+ cations, leading to stronger directional intermolecular interactions that are absent in $(\text{PMA})_2\text{PbBr}_4$.

Fig. R1. The arrangement of the PMA⁺ cations in (PMA)₂PbBr₄ (left) and (PMA)₂PbI₄ (right) viewed along the out-of-plane direction, showing the emergence of anisotropic and directional intermolecular interactions by inorganic lattice expansion.

Similar explanation can be applied to (HA)₂(MA)Pb₂I₇ versus (HA)₂(GA)Pb₂I₇ (new Fig. S20). In (HA)₂(MA)Pb₂I₇, the tails of the HA cations exhibit off-centering displacement due to anisotropic intermolecular interactions. When the inorganic lattice is expanded by incorporating larger GA⁺ cations, the off-centering displacement of the HA⁺ cation tails is further enhanced, as evidenced by the carbon–carbon distances reported in Table S3 (reproduced below). As a result, (HA)₂(GA)Pb₂I₇ exhibits greater anisotropy in the inorganic lattice compared to (HA)₂(MA)Pb₂I₇, as also reflected by the difference in the in-plane lattice parameters.

Fig. S20. Comparisons of the crystal structures and directional Van der Waals interactions of the HA⁺ cations in (HA)₂(MA)Pb₂I₇ and (HA)₂(GA)Pb₂I₇. a, b, Crystal structures of (HA)₂(MA)Pb₂I₇ (a) and (HA)₂(GA)Pb₂I₇ (b) viewed along the in-plane direction. c, d, The arrangement of the HA⁺ cations in (HA)₂(MA)Pb₂I₇ (c) and (HA)₂(GA)Pb₂I₇ (d) viewed along the out-of-plane direction. e, Optical images of (HA)₂(MA)Pb₂I₇ with elongated-shape and (HA)₂(GA)Pb₂I₇ NWs with larger aspect ratios due to stronger intermolecular interaction anisotropy. Scale bars are 25 μm.

Table S3. Comparison of the carbon-carbon distances along the in-plane shorter-axis and longer-axis between neighboring HA⁺ in (HA)₂(MA) Pb₂I₇ and (HA)₂(GA) Pb₂I₇.

Perovskite	Carbon atom	Distance along the shorter axis/Å	Distance along the longer axis/Å	Anisotropic ratio (longer/shorter)	In-plane a-axis/Å	In-plane b-axis/Å
(HA) ₂ (MA) Pb ₂ I ₇	C1	5.63	6.80	1.21	8.84	8.70
	C2	6.06	6.63	1.10		
	C3	4.85	7.96	1.64		
	C4	5.22	7.35	1.41		
	C5	4.44	9.03	2.04		
	C6	4.64	8.40	1.81		
(HA) ₂ (GA) Pb ₂ I ₇	C1	5.58/5.62 *	7.05/7.04	1.26/1.25	9.03	8.82
	C2	5.99/5.99	6.74/6.64	1.13/1.11		
	C3	4.48/4.66	8.66/8.66	1.93/1.86		
	C4	4.81/5.08	8.58/8.37	1.78/1.65		
	C5	4.07/4.46	10.67/10.63	2.62/2.38		
	C6	4.48/4.99	10.58/10.35	2.36/2.07		

* The HA⁺ cations are disordered in the (HA)₂(GA)Pb₂I₇, and thus there are two set of data. The carbon atom adjacent to the ammonium group is designated as C1, while the carbon atom at the terminus of the alkyl chain is labelled as C6.

2. Anisotropy of the inorganic lattice. We agree with the Reviewer that the growth of hybrid 2D perovskites relies on the synergy between the inorganic and organic lattices. The anisotropy of the inorganic lattice can be characterized by the difference between two in-plane lattice parameters, D₁ and D₂, (see Fig. S13 reproduced below), which represent the distances between the second-nearest lead atoms. For most 2D perovskite structures, these two values correspond to the two in-plane unit cell parameters.

As the Reviewer pointed out, the anisotropy of the inorganic lattices is indeed influenced by the anisotropic intermolecular interactions within the organic layers. When spacer cations are positioned at the center of the pockets formed by four lead halide octahedra, the inorganic lattices are isotropic along the two in-plane axes, as observed in (PEA)₂PbI₄ and (PMA)₂PbBr₄ (Fig. S13a). In contrast, if the spacer cations are displaced from the center, directional intermolecular interactions arise, introducing anisotropy into the inorganic lattice, as observed in (PMA)₂PbI₄ (Fig. S13b). Therefore, the directional intermolecular interactions are closely correlated with the anisotropy of the crystal structure. As shown in the newly added Table S2 (reproduced below), stronger

directional intermolecular interactions generally lead to tighter packing of the inorganic framework along the corresponding direction, resulting in shorter in-plane lattice parameters (i.e., typically $D_2 < D_1$). Therefore, the direction with the smaller in-plane lattice parameter often corresponds to the longer axis of the anisotropic crystal.

Fig. S13 Anisotropic inorganic lattices and anisotropic growth induced by directional intermolecular interactions. **a, b**, Schematic illustration of isotropic inorganic lattices (a) and anisotropic inorganic lattices induced by directional intermolecular interactions (b). **c**, Schematic illustration of the needle-like crystal, showing the growth direction along with the directional noncovalent intermolecular interaction.

Interestingly, we would like to point out two exceptions, $(ABA)_2PbI_4$ and $(2CF_3PEA)_2PbI_4$, which exhibit directional hydrogen bonds and $\pi \cdots \pi$ stacking interactions, respectively. However, the corresponding direction of stronger intermolecular interaction has a slightly larger in-plane lattice parameter than the other direction. Nevertheless, the crystal growth directions of these two structures align with the directional intermolecular interactions, emphasizing the critical role of these interactions in guiding 1D growth. Additionally, we note that many spacer cations are generally undersized for the lead iodide framework, as evidenced by the presence of in-plane octahedral tilting. This undersizing often results in slight off-centering displacement, and the inorganic lattices exhibit anisotropy, as observed in $(BA)_2PbI_4$. However, under our synthetic conditions, the directional anisotropic interactions in such systems are not sufficiently strong to drive significant 1D growth. Further details regarding this aspect are provided in our response to your next comment.

Table S2. Summary of in-plane lattice parameters, types and directions of intermolecular interaction chains, and crystal growth directions for various 2D perovskites.

Perovskite	CCDC	Longer-axis D ₁ (Å)	Shorter-axis D ₂ (Å)	Anisotropic ratio D ₁ /D ₂	Intermolecular interaction chain	1D growth direction
(PMA) ₂ PbI ₄	1947894	9.156	8.689	1.054	C–H···π along D ₂	along D ₂
(2FPMA) ₂ PbI ₄	2125045	9.155	8.699	1.052	C–H···π along D ₂	along D ₂
(4FPMA) ₂ PbI ₄	1947900	9.244	8.696	1.063	C–H···π along D ₂	along D ₂
(MBA) ₂ PbI ₄	1877049	9.311	8.894	1.047	C–H···π along D ₂	along D ₂
(DFP) ₂ PbI ₄	2125699	9.289	8.982	1.034	Hydrogen bond along D ₂	along D ₂
(DFPD) ₂ PbI ₄	1934899	9.346	9.011	1.037	Hydrogen bond along D ₂	along D ₂
(PA) ₂ PbI ₄	665692	8.930	8.672	1.030	CH ₃ ···CH ₃ along D ₂	along D ₂
(ABA) ₂ PbI ₄	267398	9.280	8.906	1.042	Hydrogen bond along D ₁	along D ₁
(2CF ₃ PEA) ₂ PbI ₄	2019010	8.665	8.534	1.015	π···π along D ₁	along D ₁
(PMA) ₂ SnI ₄	2375615	9.094	8.667	1.049	C–H···π along D ₂	along D ₂
(2FPMA) ₂ SnI ₄	2375611 *	9.104	8.667	1.050	C–H···π along D ₂	along D ₂
(PA) ₂ SnI ₄	2375614	8.975	8.579	1.046	CH ₃ ···CH ₃ along D ₂	along D ₂
(DFP) ₂ SnI ₄	2375612 *	9.311	8.950	1.040	Hydrogen bonds along D ₂	along D ₂
(DFPD) ₂ SnI ₄	2375613 *	9.501	8.872	1.071	Hydrogen bonds along D ₂	along D ₂
(MBA) ₂ SnI ₄	1994337	9.357	8.910	1.050	C–H···π along D ₂	along D ₂
(2CF ₃ PEA) ₂ CuCl ₄	2375609 *	8.253	7.666	1.077	π···π along D ₂	along D ₂
(CH ₂ O ₂ PEA) ₂ CdCl ₄	2375613	7.432	7.496	0.991	π···π along D ₂	along D ₂
(TAP3) ₂ PbBr ₄	2292513	9.540	7.896	1.208	π···π along D ₂ and hydrogen bond along D ₁	along D ₂
(BrCA3) ₂ PbBr ₄	2292518	8.949	7.760	1.153	π···π along D ₂	along D ₂
(BA) ₂ PbI ₄	665690	8.876	8.693	1.021	Weak CH ₃ ···CH ₃ along D ₂	Near 2D growth
(HA) ₂ PbI ₄	665695	8.941	8.687	1.029	Weak CH ₃ ···CH ₃ along D ₂	Near 2D growth
(PEA) ₂ PbI ₄	1841681	8.744	8.744	1.000	no directional non- covalent interaction	2D growth

$(4CF_3PMA)_2PbI_4$	2041929	8.691	8.418	1.032	no directional non-covalent interaction	2D growth
$(PMA)_2PbBr_4$	1542460	8.147	8.123	1.003	no directional non-covalent interaction	2D growth

* These are newly reported crystal structures. The definition of D_1 and D_2 is shown in Fig. S13.

In the revised manuscript, we have incorporated the above discussion on pages 11&12.

3. Related to comment #2, the reviewer would also like to challenge the comparison between BA, PA, and HA -based systems with similar alkylammonium backbone. While the anisotropy within spacer non-covalent interactions (a axis v.s. b) was attenuated from PA -based system to BA or HA equivalents, it's hard to justify how much attenuation would lead to disappearance of 1D growth. If CH- π interaction was the sole reason giving rise to 1D growth, then the smaller discrepancies in CH \cdots CH \cdots CH \cdots CH \cdots distances along the two in-plane directions in BA or HA-based systems might also be non-negligible. Then why they cannot grow into 1D structures, even with a fast nucleation using the droplet method?

We thank the Reviewer for raising this critical question. If the spacer cations were to occupy the exact center of the pockets formed by four lead iodide octahedra, the resulting intermolecular interactions would be isotropic, leading to the most ideal 2D growth in the proposed mechanism. However, alkylammonium cations are generally undersized relative to the lead iodide framework, leading to a slight off-centering displacement of the spacer cations in the room-temperature phases of these structures. $(PA)_2PbI_4$ is an exception, which exhibits significant larger displacements. In fact, the 2D growth of $(BA)_2PbI_4$ and $(HA)_2PbI_4$ shows minor anisotropy, as evidenced by the rectangular crystal shapes observed in some cases (see newly added Fig. S6). Nevertheless, the relatively weak directional anisotropic interactions of $(BA)_2PbI_4$ and $(HA)_2PbI_4$ are insufficient to induce significant 1D growth under our synthetic conditions.

Fig. S6. The crystal growth of $(\text{BA})_2\text{PbI}_4$ and $(\text{HA})_2\text{PbI}_4$ showing minor anisotropic growth. a, b, Optical images of some $(\text{BA})_2\text{PbI}_4$ (a) and $(\text{HA})_2\text{PbI}_4$ (b) crystals with rectangular shape. The scale bars are $25\ \mu\text{m}$. c, d, Optical image (c) and AFM image (d) showing minor anisotropic screw-dislocation cores in $(\text{BA})_2\text{PbI}_4$. The scale bars in c and d are $10\ \mu\text{m}$ and $2\ \mu\text{m}$, respectively.

The pronounced difference in the crystal growth behavior of $(\text{PA})_2\text{PbI}_4$ compared to $(\text{BA})_2\text{PbI}_4$ and $(\text{HA})_2\text{PbI}_4$ arises from differences in their structural phases and the strength of the associated intermolecular interactions. At high temperatures above 330 K, $(\text{BA})_2\text{PbI}_4$, $(\text{PA})_2\text{PbI}_4$, and $(\text{HA})_2\text{PbI}_4$ are isostructural. As the temperature decreases, the alkyl chain tails of the spacer cations deviate further from the center of the pockets (new Fig. S7), enhancing the directional intermolecular interactions that drive phase transitions in these structures. A key distinction of $(\text{PA})_2\text{PbI}_4$ is its higher phase transition temperature 319 K, which is above the temperature used for nanowire growth. Consequently, the room temperature phase of $(\text{PA})_2\text{PbI}_4$ is more closely related to the low-temperature phases of $(\text{HA})_2\text{PbI}_4$, which exhibit stronger anisotropic interactions (new Table S1). This difference accounts for the distinct growth of $(\text{PA})_2\text{PbI}_4$ compared to $(\text{BA})_2\text{PbI}_4$ and $(\text{HA})_2\text{PbI}_4$.

To further confirm this hypothesis and highlight the impact of attenuated anisotropy leading to disappearance of 1D growth, we performed crystal growth of $(\text{PA})_2\text{PbI}_4$ at 340 K, above its phase transition. At this temperature, $(\text{PA})_2\text{PbI}_4$ adopts a structure isostructural to the room temperature phases of $(\text{BA})_2\text{PbI}_4$ and $(\text{HA})_2\text{PbI}_4$, with longer distances between the tails of neighboring PA cations compared to room-temperature phase $(\text{PA})_2\text{PbI}_4$, as shown in the newly added Table S1. As expected, $(\text{PA})_2\text{PbI}_4$ grown under 340 K forms 2D plate-like crystals (new Fig. S8).

Fig. S7. Structural phases and the associated intermolecular interactions in $(PA)_2PbI_4$ and $(HA)_2PbI_4$. **a, b**, Crystal structures of high-temperature phase of $(PA)_2PbI_4$ viewed along the a -axis (a) and c -axis (b). **c, d**, Crystal structures of low-temperature phase of $(PA)_2PbI_4$ (i.e., room-temperature phase) viewed along the a -axis (c) and c -axis (d). **e, f**, Crystal structures of high-temperature phase of $(HA)_2PbI_4$ (i.e., room-temperature phase) viewed along the a -axis (e) and c -axis (f). **g, h**, Crystal structures of low-temperature phase of $(HA)_2PbI_4$ viewed along the a -axis (g) and c -axis (h).

Table S1. Comparisons of the structural phases and the associated intermolecular interactions in $(BA)_2PbI_4$, $(PA)_2PbI_4$, and $(HA)_2PbI_4$.

		$(BA)_2PbI_4$		$(PA)_2PbI_4$		$(HA)_2PbI_4$	
Phase transition temperature		274 K		319 K		268 K	
Phase		Low-T phase (223 K)	High-T phase (293 K)	Low-T phase (293 K)	High-T phase (333 K)	Low-T phase (173 K)	High-T phase (293 K)
Space group		Pbca	Pbca	P2 ₁ /a	Pbca	P2 ₁ /a	Pbca
In-plane a (Å)		8.428	8.876	8.672	9.008	8.643	8.941
In-plane b (Å)		8.986	8.692	8.930	8.731	8.845	8.687
C1	Distances along shorter / longer axis (Å)	5.37 / 7.09	5.83 / 6.61	6.63 / 6.09	5.93 / 6.63	5.74 / 6.66	5.86 / 6.63
	Anisotropic ratio*	1.32	1.13	0.91	1.12	1.16	1.13
C2	Distances along shorter / longer axis (Å)	5.69 / 6.68	5.99 / 6.44	4.79 / 8.14	6.09 / 6.46	6.06 / 6.31	6.04 / 6.43
	Anisotropic ratio	1.17	1.08	1.70	1.06	1.04	1.06

C3	Distances along shorter / longer axis (Å)	4.78 / 8.16	4.81 / 8.08	4.72 / 8.28	4.70 / 8.47	4.85 / 7.93	4.95 / 7.87
	Anisotropic ratio	1.71	1.68	1.75	1.80	1.64	1.59
C4	Distances along shorter / longer axis (Å)	5.07 / 7.56	4.97 / 7.79	4.40 / 10.62	4.94 / 8.00	5.26 / 7.31	5.27 / 7.37
	Anisotropic ratio	1.49	1.57	2.41	1.62	1.40	1.40
C5	Distances along shorter / longer axis (Å)	-	-	4.38 / 10.49	5.41 / 7.27	4.43 / 8.98	4.58 / 8.65
	Anisotropic ratio	-	-	2.39	1.34	2.03	1.89
C6	Distances along shorter / longer axis (Å)	-	-	-	-	4.65 / 8.34	4.85 / 8.05
	Anisotropic ratio	-	-	-	-	1.79	1.66

Note that the shaded columns are the room-temperature phases. The carbon atom adjacent to the ammonium group is designated as C1, while the carbon atom at the terminus of the alkyl chain is labelled as C4(BA)/C5(PA)/C6(HA). * The anisotropic ratio is defined as the ratio of the carbon-carbon distance between neighboring alkylammonium cations along the longer axis to the corresponding distance along the shorter axis.

Fig. S8. Optical images of $(PA)_2PbI_4$ crystals grown at room temperature(a) and 340K(b) reveal distinct morphologies corresponding to different structural phases. The scale bars are 2.7 mm.

Among the various types of intermolecular interactions, the van der Waals interactions between CH_3 groups ($CH_3 \cdots CH_3$) are the weakest, with interaction energies ranging from approximately 0.25 to 2 kcal/mol. For comparison, $C-H \cdots \pi$ interactions, often classified as weak hydrogen bonds, exhibit stronger interaction energies in the range of 1 to 4 kcal/mol (*Chem.* **2019**, 5, 2814). We suggest that the directional $CH_3 \cdots CH_3$ van der Waals interactions lack sufficient strength to drive 1D growth in the room-temperature phases of $(BA)_2PbI_4$ and $(HA)_2PbI_4$. However, as demonstrated in $(PA)_2PbI_4$, these interactions can become significant when further enhanced, thereby promoting anisotropic growth under suitable conditions.

In the revised manuscript, we have incorporated the above discussion on pages 8&9.

4. Thus, the reviewer believes that the authors are presenting a different growth mechanism utilizing non-covalent interactions. Yet, this mechanism did not contradict, but rather complemented the mechanism cited on page 8 using solvation effects from carboxylic acids. Particularly, the linear H-bond chain within $(\text{ABA})_2\text{PbI}_4$ differs fundamentally from the carboxylic dimer synthon. The structural flexibility of carboxylic acid -terminated alkyl chain also differs from rigid benzoic acid. Comparison across two mechanisms should be more carefully addressed, rather than merely pointing out the existence of pi-pi interaction within the cited systems. Hence, the discussions on page 8 should be revised.

The system here also introduced another merit, as they can be processed in non-aqueous solvents. The authors should expand more on the growth in non-aqueous systems to generalize the solvent effects.

We thank the Reviewer for recognizing the significance of our manuscript in presenting a distinct growth mechanism driven by non-covalent interactions. Following the Reviewer's suggestions, we have revised the corresponding discussion on pages 11&12 of the revised manuscript. Additionally, we have further investigated the crystal growth of several 2D perovskites in non-aqueous solvents. The new results, provided in new Fig. S14, demonstrate the persistent 1D crystal growth in acetonitrile (MeCN) and tetrahydrofuran (THF) through solvent evaporation.

Fig. S14. Optical images of 2D perovskite nanowires grown in organic solvent. All scale bars are 25 μm . MeCN = acetonitrile, THF = tetrahydrofuran.

Additionally, a few minor issues should be corrected:

5. Figs. 4a, S20, S29: images of the wire should come with the polar graph to show their alignment.

We have added the images of the wires. Thanks for the suggestions.

6. Fig 17d should be removed due to the unreliable comparison between data obtained from different setups.

We have removed Fig 17d.

7. Typos: S15 figure title; page 11 line 240: should be copper; page 14, line 307: should be Fig 27

We have corrected these typos. Thanks!

Reviewer #2:

The paper '2D Organic-Inorganic Hybrid Perovskite Quantum-Well Nanowires Enabled by Directional Noncovalent Intermolecular Interactions' by Mr. Zhang and co-authors, aims to demonstrate a crystal growth mechanism that utilizes directional noncovalent intermolecular interactions to synthesize 1D forms of diverse 2D metal halide perovskites. The authors synthesize different types of crystals and try to correlate the intermolecular interactions among the spacer cations with the shape of the millimeter-sized crystals.

The most serious problem is that the explanations provided by the authors are not sufficiently convincing and it is not clear which type of interaction (hydrogen bonds? van der Waals?) can promote 1D growth, there is no trend and not all the experimental findings support their hypotheses.

We appreciate the Reviewer for providing these critical comments. We apologize for the insufficient explanations and poor presentation that may have caused potential confusions. Our results demonstrate that various types of directional intermolecular interactions, including C–H \cdots π interaction, $\pi\cdots\pi$ interaction, hydrogen bond, and van der Waals CH₃ \cdots CH₃ interaction, can be unutilized to promote 1D growth. Among these interactions, the van der Waals CH₃ \cdots CH₃ interaction has the weakest strength. Consequently, in certain structures such as (BA)₂PbI₄ and (HA)₂PbI₄, the directional intermolecular interactions are not strong enough to induce significant 1D growth. However, when these interactions are further enhanced, as shown in (PA)₂PbI₄, stronger anisotropic growth can be observed. Below, we provide detailed responses to address the Reviewer's questions.

For example:

- on pages 5 and 6, it is shown that directional C–H \cdots π interactions give 1D crystals ((PMA)₂PbI₄) while isotropic C–H \cdots π interactions give 2D-plate morphology ((PMA)₂PbBr₄). However, this no longer applies in the case of BA⁺, HA⁺ and PA⁺ (p. 7) where the intermolecular interactions are anisotropic for all 3 cations but PA⁺ leads to 1D crystals while HA⁺ and BA⁺ produce 2D crystals, so directional noncovalent intermolecular interactions do not seem to play a fundamental role despite what the authors would like to prove.

We thank the Reviewer for raising this critical question. Please also refer to our responses to the comments 2 and 3 from Reviewer #1. If the spacer cations were to occupy the exact center of the pockets formed by four lead halide octahedra, the resulting

intermolecular interactions would be isotropic, leading to uniform 2D growth as proposed in our mechanism. This is observed in $(\text{PMA})_2\text{PbBr}_4$. However, due to the undersized nature of alkylammonium cations relative to the lead iodide framework, these cations typically exhibit off-centering displacements in the crystal structures, and hence present directional and anisotropic intermolecular interactions. Therefore, one would expect in-plane anisotropic growth for $(\text{BA})_2\text{PbI}_4$, $(\text{PA})_2\text{PbI}_4$, and $(\text{HA})_2\text{PbI}_4$. In fact, the 2D growth of $(\text{BA})_2\text{PbI}_4$ and $(\text{HA})_2\text{PbI}_4$ shows minor anisotropy, as evidenced by the rectangular crystal shapes observed in some cases. Additionally, slightly anisotropic screw-dislocation cores are observable under low-supersaturation growth conditions (see newly added Fig. S6). However, we argue that the directional anisotropic interactions in $(\text{BA})_2\text{PbI}_4$ and $(\text{HA})_2\text{PbI}_4$ are weaker compared to those in $(\text{PA})_2\text{PbI}_4$ and insufficient to induce significant 1D growth under our synthetic conditions, as detailed below.

Fig. S6. The crystal growth of $(\text{BA})_2\text{PbI}_4$ and $(\text{HA})_2\text{PbI}_4$ showing minor anisotropic growth. a, b, Optical images of some $(\text{BA})_2\text{PbI}_4$ (a) and $(\text{HA})_2\text{PbI}_4$ (b) crystals with rectangular shape. The scale bars are $25\ \mu\text{m}$. c, d, Optical image (c) and AFM image (d) showing minor anisotropic screw-dislocation cores in $(\text{BA})_2\text{PbI}_4$. The scale bars in c and d are $10\ \mu\text{m}$ and $2\ \mu\text{m}$, respectively.

The pronounced difference in the crystal growth behavior of $(\text{PA})_2\text{PbI}_4$ compared to $(\text{BA})_2\text{PbI}_4$ and $(\text{HA})_2\text{PbI}_4$ arises from differences in their structural phases and the strength of the associated intermolecular interactions. At high temperatures above 330K , $(\text{BA})_2\text{PbI}_4$, $(\text{PA})_2\text{PbI}_4$, and $(\text{HA})_2\text{PbI}_4$ are isostructural. As the temperature decreases, the spacer cations deviate further from the center of the pockets (Fig. S7), enhancing the directional intermolecular interactions that drive phase transitions in these structures. A key distinction of $(\text{PA})_2\text{PbI}_4$ is its higher phase transition temperature 319K , which is above the temperature used for nanowire growth. Consequently, the room temperature phase of $(\text{PA})_2\text{PbI}_4$ (space group $\text{P}2_1/\text{a}$) is more closely related to the low-temperature phases of $(\text{HA})_2\text{PbI}_4$, which exhibit stronger anisotropic interactions. This difference accounts for the distinct growth of $(\text{PA})_2\text{PbI}_4$ compared to $(\text{BA})_2\text{PbI}_4$ and $(\text{HA})_2\text{PbI}_4$.

Fig. S7. Structural phases and the associated intermolecular interactions in $(\text{PA})_2\text{PbI}_4$ and $(\text{HA})_2\text{PbI}_4$. a, b, Crystal structures of high-temperature phase of $(\text{PA})_2\text{PbI}_4$ viewed along the a -axis (a) and c -axis (b). c, d, Crystal structures of low-temperature phase of $(\text{PA})_2\text{PbI}_4$ (i.e., room-temperature phase) viewed along the a -axis (c) and c -axis (d). e, f, Crystal structures of high-temperature phase of $(\text{HA})_2\text{PbI}_4$ (i.e., room-temperature phase) viewed along the a -axis (e) and c -axis (f). g, h, Crystal structures of low-temperature phase of $(\text{HA})_2\text{PbI}_4$ viewed along the a -axis (g) and c -axis (h).

To further highlight the impact of attenuated anisotropy leading to disappearance of 1D growth, we performed crystal growth of $(\text{PA})_2\text{PbI}_4$ at 340 K, above its phase transition. At this temperature, $(\text{PA})_2\text{PbI}_4$ adopt a structure isostructural to the room temperature phases of $(\text{BA})_2\text{PbI}_4$ and $(\text{HA})_2\text{PbI}_4$, with longer distances between the tails of neighboring PA cations compared to room-temperature phase $(\text{PA})_2\text{PbI}_4$, as shown in the newly added Table S1. As expected, $(\text{PA})_2\text{PbI}_4$ grown under 340 K forms 2D plate-like crystals (new Fig. S8).

Table S1. Comparisons of the structural phases and the associated intermolecular interactions in $(\text{BA})_2\text{PbI}_4$, $(\text{PA})_2\text{PbI}_4$, and $(\text{HA})_2\text{PbI}_4$.

	$(\text{BA})_2\text{PbI}_4$		$(\text{PA})_2\text{PbI}_4$		$(\text{HA})_2\text{PbI}_4$	
Phase transition temperature	274 K		319 K		268 K	
Phase	Low-T phase (223 K)	High-T phase (293 K)	Low-T phase (293 K)	High-T phase (333 K)	Low-T phase (173 K)	High-T phase (293 K)
Space group	Pbca	Pbca	P2 ₁ /a	Pbca	P2 ₁ /a	Pbca
In-plane a (Å)	8.428	8.876	8.672	9.008	8.643	8.941

	In-plane b (Å)	8.986	8.692	8.930	8.731	8.845	8.687
C1	Distances along shorter / longer axis (Å)	5.37 / 7.09	5.83 / 6.61	6.63 / 6.09	5.93 / 6.63	5.74 / 6.66	5.86 / 6.63
	Anisotropic ratio*	1.32	1.13	0.91	1.12	1.16	1.13
C2	Distances along shorter / longer axis (Å)	5.69 / 6.68	5.99 / 6.44	4.79 / 8.14	6.09 / 6.46	6.06 / 6.31	6.04 / 6.43
	Anisotropic ratio	1.17	1.08	1.70	1.06	1.04	1.06
C3	Distances along shorter / longer axis (Å)	4.78 / 8.16	4.81 / 8.08	4.72 / 8.28	4.70 / 8.47	4.85 / 7.93	4.95 / 7.87
	Anisotropic ratio	1.71	1.68	1.75	1.80	1.64	1.59
C4	Distances along shorter / longer axis (Å)	5.07 / 7.56	4.97 / 7.79	4.40 / 10.62	4.94 / 8.00	5.26 / 7.31	5.27 / 7.37
	Anisotropic ratio	1.49	1.57	2.41	1.62	1.40	1.40
C5	Distances along shorter / longer axis (Å)	-	-	4.38 / 10.49	5.41 / 7.27	4.43 / 8.98	4.58 / 8.65
	Anisotropic ratio	-	-	2.39	1.34	2.03	1.89
C6	Distances along shorter / longer axis (Å)	-	-	-	-	4.65 / 8.34	4.85 / 8.05
	Anisotropic ratio	-	-	-	-	1.79	1.66

Note that the shaded columns are the room-temperature phases. The carbon atom adjacent to the ammonium group is designated as C1, while the carbon atom at the terminus of the alkyl chain is labelled as C4(BA)/C5(PA)/C6(HA). * The anisotropic ratio is defined as the ratio of the carbon-carbon distance between neighboring alkylammonium cations along the longer axis to the corresponding distance along the shorter axis.

Fig. S8. Optical images of $(PA)_2PbI_4$ crystals grown at room temperature(a) and 340K(b) reveal distinct morphologies corresponding to different structural phases. The scale bars are 2.7 mm.

Among the various types of intermolecular interactions, the van der Waals interactions between CH_3 groups ($CH_3 \cdots CH_3$) are the weakest, with interaction energies of 0.25-2 kcal/mol. For comparison, $C-H \cdots \pi$ interactions, $\pi \cdots \pi$ interactions, and

hydrogen bonds have larger interaction energies, typically ranging from 1-4 kcal/mol, 2-10 kcal/mol, and 5-40 kcal/mol, respectively (*Chem.* **2019**, 5, 2814). Therefore, we suggest that the directional $\text{CH}_3 \cdots \text{CH}_3$ van der Waals interactions lack sufficient strength to drive 1D growth in the room-temperature phases of $(\text{BA})_2\text{PbI}_4$ and $(\text{HA})_2\text{PbI}_4$. However, as demonstrated in $(\text{PA})_2\text{PbI}_4$, these interactions can become significant when further enhanced, thereby promoting anisotropic growth under suitable conditions.

In the revised manuscript, we have incorporated the above discussion on pages 8&9.

- at page, lines 131-134: the authors state that ‘the halogen- π interactions in 4FPMA^+ are significantly weaker than in 4CIPMA^+ ’; can they explain why, taking into account that F is more electronegative than Cl?

We apologize for the insufficient explanation provided. According to IUPAC, a halogen bond is an interaction that “occurs when there is evidence of a net attractive interaction between an electrophilic region (the σ -hole) associated with a halogen atom (the halogen-bond donor) in a molecular entity and a nucleophilic region in another, or the same, molecular entity (the halogen bond acceptor)” (*Pure Appl. Chem.* **2013**, 85, 1711). In this case, the π -system acts as the halogen bond acceptor (i.e., a Lewis base). The strength of the halogen bond is influenced by the size and depth of the σ -hole, which follows the trend: $\text{I} > \text{Br} > \text{Cl} \gg \text{F}$ (*Cryst. Growth Des.* **2019**, 19, 1426; *J. Phys. Chem. A.* **2016**, 120, 7020; *Phys. Chem. Chem. Phys.* **2010**, 12, 4543; *J. Am. Chem. Soc.* 2020, **142**, 11, 5060). Therefore, the halogen- π interactions in 4FPMA^+ are significantly weaker than in 4CIPMA^+ .

Additionally, the relative strengths of the halogen- π interactions in these systems can be observed in the crystal structures. In $(4\text{CIPMA})_2\text{PbI}_4$, the distance between chlorine and carbon atoms (3.5 Å) is close to the sum of their van der Waals radii (3.5 Å), indicating a significant interaction. In contrast, for $(4\text{FPMA})_2\text{PbI}_4$, the distance between fluorine and carbon atoms of the benzene ring (3.3 Å) is larger than the sum of their van der Waals radii (3.0 Å), suggesting a weaker interaction.

In the revised manuscript, we have incorporated the above discussion on page 7.

- on page 7, lines 135-138 it is not clear how the intermolecular interactions of $4\text{CF}_3\text{PMA}^+$ and PEA^+ can direct the crystal growth and why.

We apologize for the insufficient explanation provided. In $(\text{PEA})_2\text{PbI}_4$, the benzene rings of the PEA cations are positioned at the center of the pockets formed by four lead

iodide octahedra (Fig. R2a-c). The adjacent benzene rings do not exhibit either C–H⋯ π interactions, as seen in $(\text{PMA})_2\text{PbI}_4$, or π ⋯ π interactions, as observed in $(2\text{CF}_3\text{PEA})_2\text{PbI}_4$. Consequently, there are no directional intermolecular interaction formed along the two in-plane axes, leading to 2D growth. In $(4\text{CF}_3\text{PMA})_2\text{PbI}_4$, based on the packing arrangement (Fig. R2d-f), the adjacent benzene rings similarly lack both C–H⋯ π interactions (as in $(\text{PMA})_2\text{PbI}_4$) or π ⋯ π interactions (as seen in $(2\text{CF}_3\text{PEA})_2\text{PbI}_4$). As a result, although the benzene rings of the $4\text{CF}_3\text{PMA}$ cations are slightly displaced from the center, it is insufficient to drive significant 1D growth in $(4\text{CF}_3\text{PMA})_2\text{PbI}_4$.

For comparison, in $(2\text{CF}_3\text{PEA})_2\text{PbI}_4$, the benzene rings of the $2\text{CF}_3\text{PEA}^+$ cations exhibit a significant off-centering displacement (Fig. R2g-i). This displacement supports the formation of π ⋯ π interactions between adjacent benzene rings, evidenced by the short distances and overlapping arrangement of the rings. Thus, the lack of strong directional and anisotropic interactions, such as C–H⋯ π or π ⋯ π interactions, explains the 2D growth observed in $(4\text{CF}_3\text{PMA})_2\text{PbI}_4$ and $(\text{PEA})_2\text{PbI}_4$.

Fig R2. Crystal structures and the associated intermolecular interactions of $(4\text{CF}_3\text{PMA})_2\text{PbI}_4$, $(\text{PEA})_2\text{PbI}_4$, and $(2\text{CF}_3\text{PEA})_2\text{PbI}_4$. a-c, Crystal structures of $(\text{PEA})_2\text{PbI}_4$ viewed along the *a*-axis

(a), *b*-axis (b) and *c*-axis (c). d-f, Crystal structures of $(4CF_3PMA)_2PbI_4$ viewed along the *c*-axis (d), *b*-axis (e) and *a*-axis (f). g-i, Crystal structures of $(2CF_3PEA)_2PbI_4$ viewed along the *c*-axis (g), *b*-axis (h) and *a*-axis (i). PEA^+ = phenylethylammonium, $4CF_3PMA^+$ = 4-trifluoromethylphenylmethylammonium, and $2CF_3PEA^+$ = 2-trifluoromethylphenylethylammonium.

In the revised manuscript, we have incorporated the above discussion on page 10.

Furthermore, the introduction does not cite the relevant literature and does not mention the methods already known for obtaining 2D perovskite microwires. They only cite such literature on page 11 lines 229-235. It would be better to have a comprehensive discussion in the introduction to provide a framework for the topic of the article.

Following the Reviewer's suggestions, we have provided a more comprehensive discussion in the introduction and cited the relevant literatures on page 3.

Therefore, I do not believe that the level of novelty and soundness of this work make it suitable for publication in a high-level journal like Nature Communications.

Reviewer #3:

This work presents a universal method for growing one-dimensional (1D) perovskite single crystals using a variety of organic cation templates. It also relates to, and very similar approach with the recent findings reported in Science 384 (6699), 1000-1006, 2024. The potential impact of 1D growth of halide perovskite is not clear.

We thank the Reviewer for raising these critical comments. We would like to emphasize that our manuscript presents a fundamentally distinct mechanism for 1D anisotropic crystal growth in 2D perovskites, as also recognized by Reviewer #1. Specifically, the 1D anisotropic growth described in the Science paper is attributed to the solvation effects of carboxylic acids present in the spacer cations. In contrast, our work demonstrates the use of directional intermolecular interactions within the spacer cations to guide the crystal growth of hybrid layered perovskites.

From a crystal growth and materials engineering perspective, leveraging such directional interactions provides an effective and new strategy for controlling the morphology of hybrid materials. Given that these 2D perovskites are quantum-confined at the molecular level, the resulting 1D nanowires exhibit promising photonic properties, including significant Rabi splitting energy, low waveguiding loss coefficients, and efficient optically pumped lasing, as also noted by Reviewer #1.

In the revised manuscript, we have provided discussion to differentiate our findings from the Science report on pages 11 and 15.

Although the authors discuss growth mechanisms based on molecular interactions, additional factors, such as the intrinsic crystal structure, may influence the process but were not addressed in this paper.

We thank the Reviewer for providing this significant comment. Please also refer to our response to comment 2 from Reviewer #1. The growth of hybrid 2D perovskites requires synergy between the inorganic and organic lattices. When spacer cations are positioned at the center of the pockets formed by four lead halide octahedra, the inorganic lattices are isotropic along the two in-plane axes, as observed in $(\text{PEA})_2\text{PbI}_4$ and $(\text{PMA})_2\text{PbBr}_4$ (Fig. S13a). In contrast, if the spacer cations are displaced from the center, directional intermolecular interactions arise, introducing anisotropy into the inorganic lattice (Fig. S13b). Therefore, the directional intermolecular interactions are closely correlated with the anisotropy of the crystal structure.

Fig. S13 Anisotropic inorganic lattices and anisotropic growth induced by directional intermolecular interactions. a, b, Schematic illustration of ideal isotropic inorganic lattices (a) and anisotropic inorganic lattices (b) induced by directional intermolecular interactions. c, Schematic illustration of the needle-like crystal, showing the growth direction along with the directional noncovalent intermolecular interaction.

The anisotropy of the inorganic lattice can be characterized by the difference between two in-plane lattice parameters, D_1 and D_2 , which represent the distances between the second-nearest lead atoms (Fig. S13). As shown in the newly added Table S2 (reproduced below), stronger directional intermolecular interactions generally lead to tighter packing of the inorganic framework along the corresponding direction, resulting in a shorter in-plane lattice parameter. Therefore, the direction with the smaller in-plane lattice parameter often corresponds to the longer direction of the anisotropic crystal.

Table S2. Summary of in-plane lattice parameters, types and directions of intermolecular interaction chains, and crystal growth directions for various 2D perovskites.

Perovskite	CCDC	Longer-axis D_1 (Å)	Shorter-axis D_2 (Å)	D_1/D_2	Intermolecular interaction chain	1D growth direction
$(\text{PMA})_2\text{PbI}_4$	1947894	9.156	8.689	1.054	C–H $\cdots\pi$ along D_2	along D_2
$(2\text{FPMA})_2\text{PbI}_4$	2125045	9.155	8.699	1.052	C–H $\cdots\pi$ along D_2	along D_2
$(4\text{FPMA})_2\text{PbI}_4$	1947900	9.244	8.696	1.063	C–H $\cdots\pi$ along D_2	along D_2
$(\text{MBA})_2\text{PbI}_4$	1877049	9.311	8.894	1.047	C–H $\cdots\pi$ along D_2	along D_2
$(\text{DFP})_2\text{PbI}_4$	2125699	9.289	8.982	1.034	Hydrogen bond along D_2	along D_2
$(\text{DFPD})_2\text{PbI}_4$	1934899	9.346	9.011	1.037	Hydrogen bond along D_2	along D_2
$(\text{PA})_2\text{PbI}_4$	665692	8.930	8.672	1.030	CH $_3\cdots$ CH $_3$ along D_2	along D_2
$(\text{ABA})_2\text{PbI}_4$	267398	9.280	8.906	1.042	Hydrogen bond along D_1	along D_1

(2CF ₃ PEA) ₂ PbI ₄	2019010	8.665	8.534	1.015	$\pi \cdots \pi$ along D ₁	along D ₁
(PMA) ₂ SnI ₄	2375615	9.094	8.667	1.049	C–H \cdots π along D ₂	along D ₂
(2FPMA) ₂ SnI ₄	2375611 *	9.104	8.667	1.050	C–H \cdots π along D ₂	along D ₂
(PA) ₂ SnI ₄	2375614	8.975	8.579	1.046	CH ₃ \cdots CH ₃ along D ₂	along D ₂
(DFP) ₂ SnI ₄	2375612 *	9.311	8.950	1.040	Hydrogen bonds along D ₂	along D ₂
(DFPD) ₂ SnI ₄	2375613 *	9.501	8.872	1.071	Hydrogen bonds along D ₂	along D ₂
(MBA) ₂ SnI ₄	1994337	9.357	8.910	1.050	C–H \cdots π along D ₂	along D ₂
(2CF ₃ PEA) ₂ CuCl ₄	2375609 *	8.253	7.666	1.077	$\pi \cdots \pi$ along D ₂	along D ₂
(CH ₂ O ₂ PEA) ₂ CdCl ₄	2375613	7.432	7.496	0.991	$\pi \cdots \pi$ along D ₂	along D ₂
(TAP3) ₂ PbBr ₄	2292513	9.540	7.896	1.208	$\pi \cdots \pi$ along D ₂ and hydrogen bond along D ₁	along D ₂
(BrCA3) ₂ PbBr ₄	2292518	8.949	7.760	1.153	$\pi \cdots \pi$ along D ₂	along D ₂
(BA) ₂ PbI ₄	665690	8.876	8.693	1.021	Weak CH ₃ \cdots CH ₃ along D ₂	Near 2D growth
(HA) ₂ PbI ₄	665695	8.941	8.687	1.029	Weak CH ₃ \cdots CH ₃ along D ₂	Near 2D growth
(PEA) ₂ PbI ₄	1841681	8.744	8.744	1.000	no directional non- covalent interaction	2D growth
(4CF ₃ PMA) ₂ PbI ₄	2041929	8.691	8.418	1.032	no directional non- covalent interaction	2D growth
(PMA) ₂ PbBr ₄	1542460	8.147	8.123	1.003	no directional non- covalent interaction	2D growth

* These are newly reported crystal structures. The definition of D₁ and D₂ is shown in Fig. S13.

Interestingly, we would like to point out two exceptions, (ABA)₂PbI₄ and (2CF₃PEA)₂PbI₄, which exhibit directional hydrogen bonds and $\pi \cdots \pi$ interactions, respectively. However, the corresponding direction of stronger intermolecular interaction has a slightly larger in-plane lattice parameter than the other direction. Nevertheless, the crystal growth directions of these two structures align with the directional intermolecular interactions, emphasizing the critical role of these interactions in guiding 1D growth. Additionally, we note that 2D lead iodide perovskites such as (BA)₂PbI₄ and (HA)₂PbI₄ exhibit slightly off-center displacements, as the spacer cations are generally undersized for the lead iodide framework. Therefore, these structures also exhibit anisotropic

inorganic lattices. However, the directional anisotropic interactions in these cases are not sufficiently strong to induce significant 1D growth under our synthetic conditions. For further details, please refer to our responses to comment 3 from Reviewer #1 and comment 1 from Reviewer #2.

In the revised manuscript, we have incorporated the above discussion on pages 11&12.

The authors mention that strong hydrogen bonding and aromatic π - π stacking interactions promote 1D anisotropic growth; however, these effects also impact the dimensionality (1D, 2D, or 0D) of the resulting crystal structure. The authors should report all the single crystal structure (cif) files on the materials presented in this paper.

Following the Reviewer's suggestions, we have provided all the CIF files and confirmed the formation of 2D perovskites under our synthetic conditions (see Table S2, Fig. S17 and Supplementary cif files). Among these structures, $(\text{PMA})_2\text{PbI}_4$, $(2\text{FPMA})_2\text{PbI}_4$, $(4\text{FPMA})_2\text{PbI}_4$, $(\text{PA})_2\text{PbI}_4$, $(\text{ABA})_2\text{PbI}_4$, $(\text{DFP})_2\text{PbI}_4$, $(\text{DFPD})_2\text{PbI}_4$, $(\text{MBA})_2\text{PbI}_4$, $(2\text{CF}_3\text{PEA})_2\text{PbI}_4$, $(2\text{BrPEA})_2\text{PbI}_4$, $(\text{PMA})_2(\text{MA})\text{Pb}_2\text{I}_7$, $(\text{HA})_2(\text{GA})\text{Pb}_2\text{I}_7$, $(\text{PA})_2(\text{ATA})\text{Pb}_2\text{I}_7$, and $(\text{PMA})_2(\text{MA})_2\text{Pb}_3\text{I}_{10}$ have been previously reported in the literature, with their corresponding CCDC numbers being 1947894, 2125045, 1947900, 665692, 267398, 2125699, 1934899, 1877049, 2019010, 2000017, 1876243, 1888368, 2268600 and 1876244, respectively. We conducted single-crystal X-ray diffraction, powder X-ray diffraction, and PL measurements to confirm that the products are indeed in the 2D perovskite phase (Fig. S17). In addition, $(\text{CH}_2\text{O}_2\text{PEA})_2\text{CdCl}_4$, $(\text{DFPD})_2\text{SnI}_4$, $(\text{DFP})_2\text{SnI}_4$, $(2\text{FPMA})_2\text{SnI}_4$, and $(2\text{CF}_3\text{PEA})_2\text{CuCl}_4$ are newly reported structures in this study. We have solved their crystal structures and provided the corresponding CCDC numbers in the newly added Table S2.

Fig. S17. Powder X-ray diffraction patterns and PL spectra of various nanowires confirming the 2D perovskite phases.

As the crystals are relatively thin, the authors need to report the luminescence stability of the materials to future validate the potential for light emitter applications.

Following the Reviewer's suggestion, we have conducted experiments to evaluate the luminescence stability of the materials (see discussion on page 15 of revised manuscript and Fig. S25). Specifically, we compared several representative 2D perovskite NWs with that of the 3D perovskite MAPbI₃ under various conditions, including exposure to air, thermal heating, and UV-irradiation. The results demonstrate that 2D perovskite NWs exhibit superior luminescent stability under air and thermal conditions

compared to their 3D counterparts. Under UV-irradiation, the luminescent stability of 2D perovskite NWs with $n = 3$ is comparable to that of 3D perovskite NWs. The results have been included as Fig. S25. As noted by the Reviewer, stability remains a significant challenge for the practical application of 2D perovskites in this field. Previous studies have shown that coating 2D perovskites with polymethyl methacrylate (PMMA) offers a potential strategy to enhance their luminescence stability (ACS. Appl. Electron. Mater. 2021, 3, 1572; Science, 2024, 384, 1000).

Fig. S25. Comparisons of air stability, thermal stability, and photostability between 3D perovskite NWs and 2D perovskite NWs. The left panel shows the plot of PL intensity versus time. The right panel shows the PL image of the samples before and after degradation. All scale bars are 5 μm . a, Air stability under room temperature and humidity 76%. b, Thermal stability under 60 $^{\circ}\text{C}$ and humidity 43% in air. c, Photostability under 400 ± 5 nm light illumination from a mercury lamp and humidity 43% in air. The light power density is 0.014 W/cm^2 .

Minor comment: On page 2, the authors wrote, 'semiconductor nanowires have long fascinated scientist.' This should be revised.

We have removed the sentence.

Response to the Reviewers' Comments

We sincerely thank the Reviewers for their time and effort re-reviewing our manuscript. We have performed new experiments to address their remaining questions. To facilitate the re-evaluation process, we provide a detailed point-by-point response to the Reviewers comments, where the original comments are reproduced in black font and our corresponding responses are provided in blue. Additionally, we include an annotated version of the revised manuscript highlighting all changes.

Reviewer #1:

The authors have done an excellent job addressing the previous comments. In particular, the discussions related to comments #2 and #3 are sophisticated and comprehensive, significantly enhancing the general audience's understanding of the crystal structures and growth dynamics of layered perovskites.

As mentioned previously, the synergy between organic and inorganic components is crucial, and their interdependence often resembles a "chicken-and-egg" scenario. Regarding intermolecular interactions, Wulff construction is typically effective for explaining surface energy anisotropy. However, it becomes somewhat non-intuitive when attempting to generalize it to inorganic lattices.

We sincerely appreciate the Reviewer's recognition of our efforts in revising the manuscript. When a bulk material is cleaved into two pieces, the energy required for this process is equivalent to the total surface energy of the two new surfaces created. The overall surface energy is the combined contribution of both the organic and inorganic lattices. As for the inorganic lattices, a denser atomic packing generally indicates stronger chemical bonding in the bulk, which empirically results in a lower surface energy of the corresponding facet along the denser direction. As shown in Table 1, the directional intermolecular interactions typically align with the more denser packing directions of the inorganic lattices. Consequently, the facet perpendicular to these directional intermolecular chains is expected to have a higher overall surface energy and, therefore, a faster growth rate.

On page 12 of the revised manuscript, we have revised the sentence to more accurately describe the synergistic relationship between the organic and inorganic lattices and have further clarified our rationale for explaining the surface energy of the inorganic lattice.

A few follow-up comments:

1) As the discussions on crystal structures have now become quite comprehensive, it may be worth considering how to better present this information in the main text to enhance readability for a general audience. Additionally, please evaluate if any rearrangements or adjustments in structure might improve the flow and accessibility of the content.

We agree with the Reviewer's suggestion. In the revised manuscript, we have included new Table 1 (original Table S2) and Fig. 3 (original Fig. S13) in the main text to enhance the readability and accessibility of the content. Thank you for the advice.

2) In Fig. S14, a few candidates grew into multiple branches from organic solvents (usually 4 branches). What is the mechanism of branching? For those with a cross shape, would each branch follow the same crystallographic direction or different? (i.e. all along a in MBA_2PbI_4 or along a and b ?)

We appreciate the Reviewer's careful reading of our manuscript. The branching observed in certain structures arises due to the rapid evaporation of the organic solvent. Such fast or abrupt evaporation leads to a higher degree of supersaturation, promoting the formation of additional nucleation sites for crystal growth. As demonstrated in Fig. R1a-d, by slowing down the evaporation rate of the organic solvent, we could effectively mitigate branching in these structures. We apologize that the optimization of crystal growth in the organic solvent was not obtained in our initial revision. We have now incorporated the updated results in Fig. S14 and explained the branching in the caption.

Fig. R1. Branching crystal growth in certain 2D perovskites. (a, b), Optical images of $(\text{MBA})_2\text{PbI}_4$ crystals grown under fast evaporation (a) and slow evaporation (b) conditions. The Scale bars are $25\ \mu\text{m}$. (c, d), Optical images of $(\text{PMA})_2\text{PbI}_4$ crystals grown under fast evaporation (a) and slow evaporation (b) conditions. The Scale bars are $25\ \mu\text{m}$. (e, f), Optical images of $(\text{MBA})_2\text{PbI}_4$ (e) and $(\text{PMA})_2\text{PbI}_4$ (f) crystals with 4 branches grown in aqueous solution. The Scale bars are $50\ \mu\text{m}$. (g, h), SEM images of $(\text{MBA})_2\text{PbI}_4$ crystals with 4 branches. The Scale bars are $10\ \mu\text{m}$.

The Reviewer's curiosity regarding the growth direction of the cross-shaped structures is very inspiring. A closer examination of optical images reveals the presence of twinning at the branched center (Fig. R1a). In fact, similar cross-shaped structures have been observed in aqueous solution growth, particularly when a hotter precursor solution is dispensed onto a glass substrate, where comparable twinning at the branched center can be seen (Fig. R1e-g). Since the structures grown in organic solvents are too small for single-crystal diffraction analysis, we instead analyzed larger crystals obtained from aqueous solution growth. To determine the crystallographic growth direction, we cleaved the branches and performed single-crystal X-ray diffraction on each, as illustrated in Fig. R2. The results indicate that each branch follows the same crystallographic direction, corresponding to stronger intermolecular interactions. **These new results have been added on pages 5 & 6 as a new Fig. S3 in the revised manuscript.**

Fig. R2. Optical images of the cleaved $(\text{MBA})_2\text{PbI}_4$ branches mounted on the diffractor, confirming that each branch follows the same crystallographic direction, corresponding to stronger intermolecular interactions.

3) Please explain the side peaks in multiple PXRD spectra in Fig. S17.

We appreciate the Reviewer's careful reading of our manuscript. In the original revised manuscript, there are minor side peaks in the PXRD spectra of $(2\text{FPMA})_2\text{PbI}_4$, $(2\text{BrPEA})_2\text{PbI}_4$, and $(2\text{CF}_3\text{PEA})_2\text{PbI}_4$. As the precursor solution droplets may dry in some region of glass slides, impurities formed and were unintentionally captured for the PXRD

measurements. However, these can be avoided. We have re-performed the PXRD measurements on these structures and obtained pure spectra without side peaks. In the revised manuscript, we have updated the new results in Fig. S17, which is reproduced below.

Fig. S17. Powder X-ray diffraction patterns and PL spectra of various lead iodide phase nanowires confirming the 2D perovskite phases.

4) page 18, “Additional lasing studies of other NWs are provided in Fig. S17” – Fig number was not correct. Please check other references as well.

Thank you for pointing out the typo. We have corrected the number and checked throughout the manuscript.

5) “Generally, 2D perovskite NWs demonstrate superior air and thermal stability, whereas the photoluminescence stability of 3D perovskite NWs.” This statement seems to contradict with the results shown in Fig. S25. At least from the particles investigated,

compared with MAPbI₃, photostability was generally worse, while air and thermal stability comparable.

Based on the results presented in Fig. S25, we argue that in the air stability test with humidity 76%, 3D MAPbI₃ NWs exhibit the most significant degradation. In term of thermal stability, (PMA)₂FAPb₂I₇ and (PMA)₂(MA)Pb₂I₇ exhibits better stability than MAPbI₃. For photostability, (PMA)₂(MA)₂Pb₃I₁₀ is comparable to MAPbI₃. However, we agree that these differences are not substantial and overemphasizing them would be inappropriate. On page 17 of the revised manuscript, we have modified the sentence to "Compared to 3D MAPbI₃ NWs, 2D perovskite NWs can exhibit comparable air, thermal, and photoluminescence stability, which also vary depending on their compositions".

Reviewer #2:

The authors in the revised version have added experimental results and further discussion that enriched the work. However, in my opinion, the claim to demonstrate 'an alternative, universal mechanism for the formation of single-crystal NWs of 2D perovskite' remains unproven.

Specifically, the correlation between intermolecular interaction and crystal shape is still elusive. Even in the revised version, no clear trend or molecular design of the organic cation is presented that could predict the formation of nanowires.

We respectfully disagree with the Reviewer's opinion. As summarized in Table 1, which presents the crystal growth directions and structures of 24 different 2D perovskites, there is a strong correlation between anisotropic growth and the presence of directional intermolecular interactions. These 2D perovskites tend to grow more rapidly along directions with stronger intermolecular interactions, resulting in macroscopic elongated crystal morphologies. Subsequent crystal growth engineering is then applied to synthesize nanowires. We believe these results provide sufficient evidence to support our conclusion.

Building upon our findings, we can predict the likelihood of a given structure forming nanowires based on its crystal structure using our synthetic approach. This capability, in part, explains our success in establishing a broad library of 2D perovskite NWs incorporating diverse spacer cations. However, if the reviewer is referring to predicting whether a specific spacer cation would induce directional intermolecular interactions or generalizing its molecular motif, this remains highly challenging. The formation of

directional interactions is not solely determined by the molecular structure of the spacer cation. As shown in Fig. 2 and Table 1, a wide range of noncovalent interactions can induce directional interactions, yet these interactions are highly sensitive to even minor modifications in the spacer cation, such as a single-carbon difference in an aliphatic chain. Moreover, even when using the same spacer cation (e.g., PMA⁺), the bromide-based structure exhibits different intermolecular interactions compared to its iodide counterpart. While we agree that developing molecular design principles for spacer cations is of great interest, this is beyond the scope of the present manuscript. **We have added a discussion on this perspective on page 20.**

Indeed, in the case of (BA)₂PbI₄, (PA)₂PbI₄ and (HA)₂PbI₄, which exhibit very similar organic chains and CH₃–CH₃ van der Waals interactions, the synthesis must be conducted at significantly different temperatures (RT vs 67 °C) to achieve the assumed trend proposed in the manuscript. This discrepancy suggests that additional factors, possibly complementary to intermolecular interactions, might influence crystal shape, particularly when growth is conducted under identical conditions.

Thank you for the additional comment. In fact, only the high-temperature phase of (PA)₂PbI₄ exhibits CH₃–CH₃ van der Waals interactions and crystal structures similar to the room-temperature phases of (BA)₂PbI₄ and (HA)₂PbI₄. The room-temperature phase of (PA)₂PbI₄ exhibits significantly enhanced intermolecular interactions compared to both its high-temperature phase and the room-temperature phases of (BA)₂PbI₄ and (HA)₂PbI₄ (Fig. S8). We believe that this distinction mainly accounts for the differences in their growth behaviors.

Fig S8. Structural phases and the associated intermolecular interactions in (PA)₂PbI₄ and

(HA)₂PbI₄. a, b, Crystal structures of high-temperature phase of (PA)₂PbI₄ viewed along the a-axis (a) and c-axis (b). c, d, Crystal structures of low-temperature phase of (PA)₂PbI₄ (i.e., room-temperature phase) viewed along the a-axis (c) and c-axis (d). e, f, Crystal structures of high-temperature phase of (HA)₂PbI₄ (i.e., room-temperature phase) viewed along the a-axis (e) and c-axis (f). g, h, Crystal structures of low-temperature phase of (HA)₂PbI₄ viewed along the a-axis (g) and c-axis (h).

To further validate that the morphological transition of (PA)₂PbI₄ crystals from plate-like to needle-like at elevated temperatures is driven by the phase transition to a structure with weaker intermolecular interactions, we conducted crystal growth experiments on various 2D perovskites at 67 °C. Since these structures do not undergo phase transitions, they exhibit similar growth behaviors to those observed at room temperature. These findings suggest that strong directional intermolecular interactions predominantly influence the crystal shape, rather than the growth temperature. In the revised manuscript, we have added these new results in Fig. S9 and included the above discussion in the caption.

Fig. S9. Optical images of various 2D perovskites at room-temperature and 340 K. All Scale bares are 2.7 mm

For these reasons, I believe the paper is not suitable for publication in Nature Communications.

Reviewer #3:

The authors answered most of the questions in the response letter, I do not have further questions.

Thank you very much for the support of our manuscript.

Response to the Reviewers' Comments

Reviewer #1:

The authors have addressed all the comments very comprehensively. I appreciate the additional efforts to elucidate the branching mechanism.

Thank you very much for the support of our manuscript.

Reviewer #2:

The manuscript has significantly improved compared to the initial submission. However, I believe that framing the article as presenting a "universal mechanism for obtaining single-crystal NWs of 2D perovskites" is not entirely correct. The study does not appear to introduce a new synthetic approach; rather, it involves a screening of various organic cations, many of which lead to unidirectional growth. The findings suggest that this growth behavior is linked to molecular interactions, which vary widely and include numerous exceptions. While the screening and XRD analysis provide valuable insights, I do not find the level of novelty sufficient for publication in Nature Communications.

We sincerely appreciate the Reviewer's efforts in reviewing our manuscript, which have significantly improved our work. However, we disagree with the assertion that "...include numerous exceptions". Our work presents strong evidence for the correlation between anisotropic growth and the directional intermolecular interactions in 2D perovskites, and this correlation holds consistently at least for all the screened structures in this work, without any notable exceptions.